# A SNX1–SNX2–VAPB partnership regulates endosomal membrane rewiring in response to nutritional stress

Juliane Da Graça ⬤, Juliette Charles, Morgane Djebar, Karla Alvarez-Valadez, Joëlle Botti, Etienne Morel ⬤

**Nutrient deprivation ("starvation") is a major catabolic stress faced by mammalian cells in both pathological and physiological situations. Starvation induces autophagosome biogenesis in the immediate vicinity of ER and leads to lysosome spatial repositioning, but little is known about the consequences of nutritional stress on endosomes. Here, we report that starvation induces tethering of endosomal tubules to ER subregions, fostering autophagosome assembly. We show that this endosomal membrane generation is regulated by sorting nexin 1 (SNX1) protein and is important for the autophagic response. These newly formed SNX1 endosomal tubules establish connections with ER subdomains engaged in early autophagic machinery mobilization. Such endosome-ER transient tethers are regulated by a local dialog between SNX2, an endosomal partner of SNX1, and VAPB, an ER protein associated with autophagy initiation stage regulation. We propose that in a very early response to starvation, SNX1 and SNX2 cooperation induces and regulates endosomal membrane tubulation towards VAPB-positive ER subdomains involved in autophagosome biogenesis, highlighting the contribution of early endosomes in the cellular response to nutritional stress.**

## Introduction

Upon nutrient deprivation, mammalian cells undergo a series of signaling and membrane remodeling events, mainly building up an autophagic response to stress. Lysosomal repositioning towards perinuclear areas has been reported to occur depending on nutrient availability (Korolchuk et al, 2011; Li et al, 2016; Zhao et al, 2020), a process that is associated with mTOR mobilization and lysosomal membrane recycling via PI4,5P2 (Rong et al, 2012) and sorting nexin (SNX) proteins SNX4, SNX5, and SNX7 (Zhou et al, 2022). The most striking consequence of starvation is de novo biogenesis of the autophagosome, a double membrane vesicle aimed to capture cargoes in the cytoplasm and ensure their transport to the lysosome for destruction to promote cargos components renewal and recycling (Boya et al, 2013). Autophagosome assembly is a complex process which requires the synchronized action of specialized proteins including the ATG protein family and membrane reservoirs (Melia et al, 2020) such as ER-mediated contact sites, Golgi vesicles, mitochondria, and endosomes (Molino et al, 2017; Morel, 2020).

Assembly of autophagosomes is initiated at omegasomes, which are transient areas of the ER membrane enriched in phosphatidylinositol-3-phosphate (PI3P), a lipid synthesized by the VPS34 kinase in response to stress. The spatio-temporal coordination of PI3P-positive membranes and PI3P-binding proteins such as DCFP1 (also referred to as ZFYVE1) and WIPI2 allows the recruitment of specific ATG proteins, such as ATG16L1, ATG12, and ATG5, which leads to local lipidation of ATG8/LC3 on pre-autophagosomal membrane. This is considered essential for autophagosomal compartment identification (Melia et al, 2020). Membrane contact sites and specialized proteins such as ATG2 and ATG9A will favor the emergence of the phagophore, a double membrane cup-shaped structure known as the non-mature autophagosome that will eventually close up after capturing autophagy-related cargoes (Hurley & Young, 2017).

Several early and recycling endosome-associated proteins have been identified as key players for autophagosome assembly over the past 10 yr (Longatti & Tooze, 2012; Puri et al, 2013, 2018; Dooley et al, 2014; Wilson et al, 2014; Judith et al, 2019). However, the contribution of the endosomal membranes per se to the phagophore assembly and to its membrane fueling is still poorly understood. The endosomal protein family of SNXs plays a role in endomembrane shaping, remodeling, and transport through a phosphoinositide-binding domain (such as phox/PX domain) and, for a subset of the SNXs known as SNX-BAR, an additional C-terminal bin, amphiphysin, and Rvs (BAR) domain for membrane curvature sensing and formation (Cullen, 2008; Johannes & Wunder, 2011; Gallon & Cullen, 2015). Recently, several members of the SNXs family have been implicated in autophagic-related activities, such as delivering ATG16L1-positive membranes to pre-autophagosomal platforms (SNX18 [Knævelsrud et al, 2013; Søreng et al, 2018]) or coordinating

Université Paris Cité, INSERM UMR-S1151, CNRS UMR-S8253, Institut Necker Enfants Malades, Paris, France

Correspondence: etienne.morel@inserm.fr
Morgane Djebar's present address is Institut de Biologie Paris Seine, CNRS UMR 7622, Sorbonne Université, Paris, France
Karla Alvarez-Valadez's present address is Centre de Recherche des Cordeliers, INSERM UMR 1138, Sorbonne Université, Université de Paris, Paris, France; Metabolomics and Cell Biology Platforms, Institut Gustave Roussy, Villejuif, France

ATG9A vesicular trafficking (SNX4 and SNX7 [Antón et al, 2020; Ravussin et al, 2021]). SNX5 was shown to regulate the activity of the PI3KC3 complex at the endosomal surface of virus-infected cells to enable local autophagy initiation during viral infection (Dong et al, 2020). SNX4, SNX5, and SNX7 were reported to participate in auto-lysosomes membrane renewal (Zhou et al, 2022). The PX- and BAR-positive SNX1 protein associates with endosomal membranes to initiate formation of tubules required for retromer-associated transport from endosomes to the TGN (Johannes & Wunder, 2011; Gallon & Cullen, 2015), in a PI3P dependent manner, but it was not reported to participate in the regulation of autophagic processes so far.

In the current study, we postulate that a subset of endosomal membranes is generated and redistributed to the vicinity of ER subdomains where phagophore biogenesis is to take place in response to starvation. We show that starvation-induced autophagy associates with an increase in endosomal tubulation and net-working, a dynamic process dependent on the endosomal protein SNX1. Whereas overexpression of SNX1 increases autophagosome numbers, its knockdown is associated with reduced autophagic activity. We report that upon starvation, SNX2, an endosomal partner of SNX1, patterns to SNX1-positive tubules and triggers membrane tethering between the SNX1 endosomal tubules and autophagosome-initiating ER platforms via the ER protein VAPB. Moreover, we show that SNX2–VAPB interaction is regulated by starvation. This study highlights the importance of membrane cooperation during stress response and uncovers a new function of the SNX protein family upon stress.

# Results

### Starvation induces SNX1-positive endosomal membranes

To decipher whether starvation, which induces autophagy, impacts endosomal membranes, we analyzed the morphology of early endosomes at different time points of starvation by monitoring the early endosomal marker EEA1 (Fig 1A). Using organelle's morphology parameters such as roundness (Fig 1B) and elongation (Fig 1C), we show that early endosomes displayed an increasing number of unusual elongated structures after starvation. To test whether this altered morphology was linked to SNX1 protein, an essential regulator of early endosome tubulation (Cullen, 2008), we quantified levels of SNX1 upon autophagy induction, and we observed that the amount of SNX1 transiently increased after 1 h of starvation, after induction of autophagy, as assessed by LC3II levels (Fig S1A and B). Moreover, the overall codistribution of SNX1 and EEA1, as observed by immunofluorescence, decreased compared with the total signal of SNX1 (Fig S1C and D), suggesting that a specialization of endosomal domains might occur during the early steps of autophagy with a mobilization of SNX1-associated endosomal membranes.

We then transiently transfected HeLa cells with a GFP-tagged version of SNX1 to monitor the dynamics of SNX1-associated endosomes and to maximize the detection of tubules, which are barely detectable by light microscopy when using antibodies. We

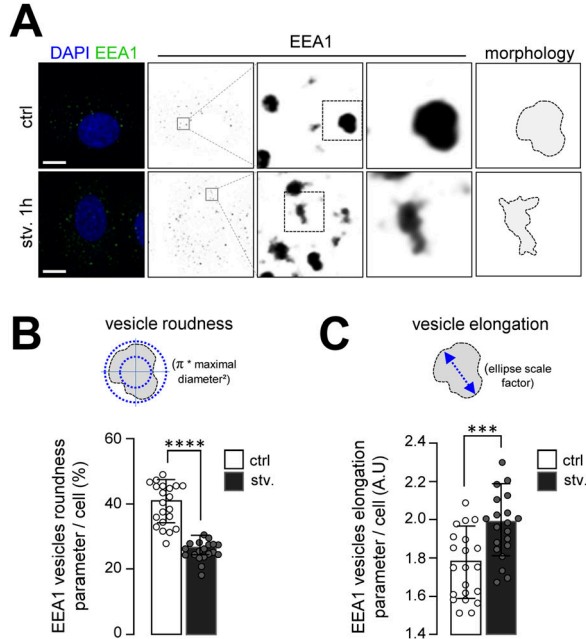

**Figure 1. Starvation promotes early endosome morphology alteration.**
**(A)** HeLa cells were grown under control and starvation for 1 h (stv. 1 h) conditions, fixed and stained with anti-EEA1 antibody (green channel) and DAPI (blue channel). Magnified areas show EEA1 endosomes in depicted conditions, with illustrated corresponding morphology (right panel). Scale bars = 10 μm. **(B)** Schematic representation of vesicle roundness parameter obtained with Icy software ROI analyses and graphical quantification of the percentage of EEA1 vesicle roundness per cell in control and starvation for 1 h (stv.) conditions. Means ± SD, from three independent experiments; ****$P < 0.0001$ in unpaired two-tailed $t$ test. **(C)** Schematic representation of vesicle elongation parameter obtained with Icy ROIs analyses and quantification of the EEA1 vesicle elongation per cell in control and starvation for 1 h (stv.) conditions. Means ± SD, from three independent experiments; ***$P < 0.001$ in unpaired two-tailed $t$ test.

observed that SNX1-GFP–positive endosomes form a tubular network under starvation, as assessed by the high increase in triple tubular connection of SNX1-GFP structures after 1 h of starvation (Fig 2A and B). The SNX1-GFP network observed in response to starvation connects with EEA1-positive endosomes (Fig 2C), indicating that SNX1 still associates with bona fide early endosomes. We show that although exogenous SNX1-GFP alters EEA1 endosome morphology by itself, this phenomenon is significantly amplified by starvation (roundness measurements, Fig S2A and B), arguing for a bona fide effect of nutrient deprivation on early endosome morphology. Live microscopy-based quantifications of SNX1-GFP–positive tubules at very short times after autophagy induction (1 and 15 min starvation, Fig 2D) revealed a significant increase in tubule number (Fig 2E) and length (Fig 2F), confirming that starvation induces SNX1 endosome tubulation early on. To ensure that the observed phenotype was not due to differences in SNX1-GFP expression among cells, we monitored the live tubulation of SNX1-positive endosomal structures in a determined acquisition field. As shown in Fig S2C and D (in HeLa cells) and in S2E and S2F (in MEF cells), short periods of starvation, such as 2 and 5 min, are sufficient to induce an increase of SNX1-associated tubules. Finally we show that Rapamycin, an inhibitor of mTOR signaling associated with nutrient sensing, drives similarly SNX1 endosome tubulation

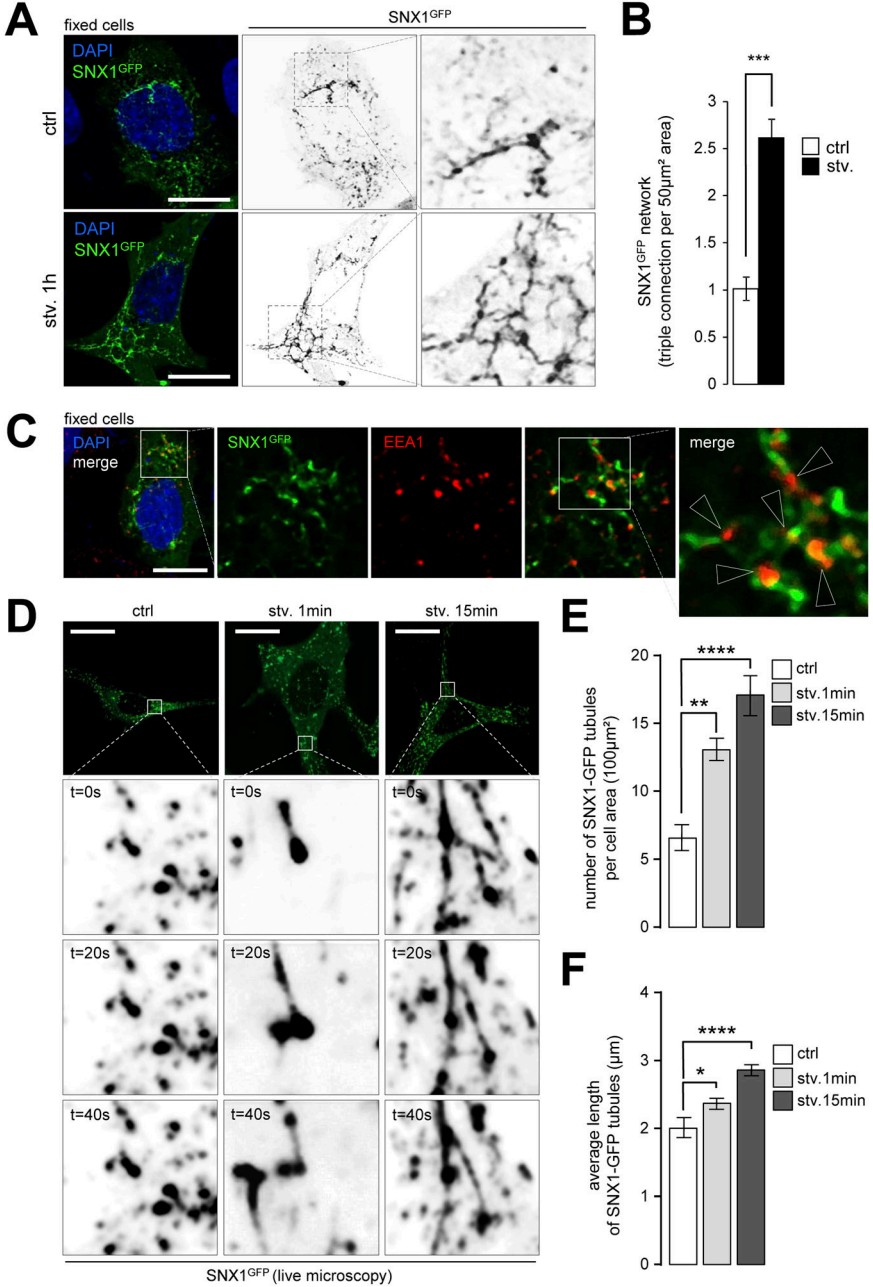

**Figure 2. SNX1-positive endosomes tubulation increases with autophagy initiation.**
**(A)** HeLa cells were transfected with SNX1-GFP, then grown under control or starvation (stv. 1 h) conditions and fixed. Magnified areas show SNX1-GFP tubular network in depicted conditions. Scale bars = 10 $\mu$m.
**(B)** Quantification of triple connection of SNX1-GFP tubules per 50 $\mu$m$^2$ forming in HeLa cells under control or starvation (stv., 1 h). Means ± SD, from three independent experiments; ***$P < 0.001$ in unpaired two-tailed $t$ test.
**(C)** HeLa cells were transfected with SNX1-GFP, grown under starvation for 1 h, fixed and stained with anti-EEA1 antibodies (red channel) and DAPI (blue channel). Scale bar = 10 $\mu$m. **(D)** Time-lapse images of HeLa cells expressing SNX1-GFP under starved conditions (stv. 1 and 15 min). Enlarged crops show SNX1-GFP–positive tubules. Scale bars = 10 $\mu$m. **(D, E, F)** Quantifications of SNX1-GFP tubules number per 100 $\mu$m$^2$ (E) and length in $\mu$m (F) as shown in (D). Means ± SEM, from three independent experiments; *$P < 0.05$, **$P < 0.01$, and ****$P < 0.0001$ in ordinary one-way ANOVA test.

(Fig S2G and H). Altogether, our data indicate that upon nutritional stress, early endosomes undergo rapid membrane morphology alterations via a local SNX1-dependent tubulation.

### Modulating SNX1 levels alter the autophagic response

Starvation and rapamycin-mediated mTOR inhibition are classical inducers of canonical autophagy (Boya et al, 2013). To address the putative effect of SNX1 endosome morphology alteration on starvation-induced autophagy, we first analyzed the behavior of autophagy marker LC3 in cells knocked down for SNX1 (Fig S3A and B) by immunofluorescence (puncta) and by Western blot (lipidation levels [i.e., LC3.II]). Upon starvation, we observed a significant decrease of LC3-positive structures corresponding to pre-autophagosomes, mature autophagosomes, and autolysosomes (Fig 3A and B). Importantly, LC3 puncta decreased also in starved cells treated with bafilomycin A1 (Baf A1, a drug used to inhibit lysosomal maturation and autophagosome–lysosome fusion), revealing an impaired autophagic flux in siSNX1 cells (Fig 3A and B, +Baf A1 conditions), as confirmed by Western blot analyses of LC3.II levels in similar conditions (Fig S3C and D), suggesting that reducing SNX1 levels alters autophagosome biogenesis. This was further confirmed by fluorescence-based analysis of the cellular distribution of autophagy early stages markers ATG16L1 and WIPI2, two

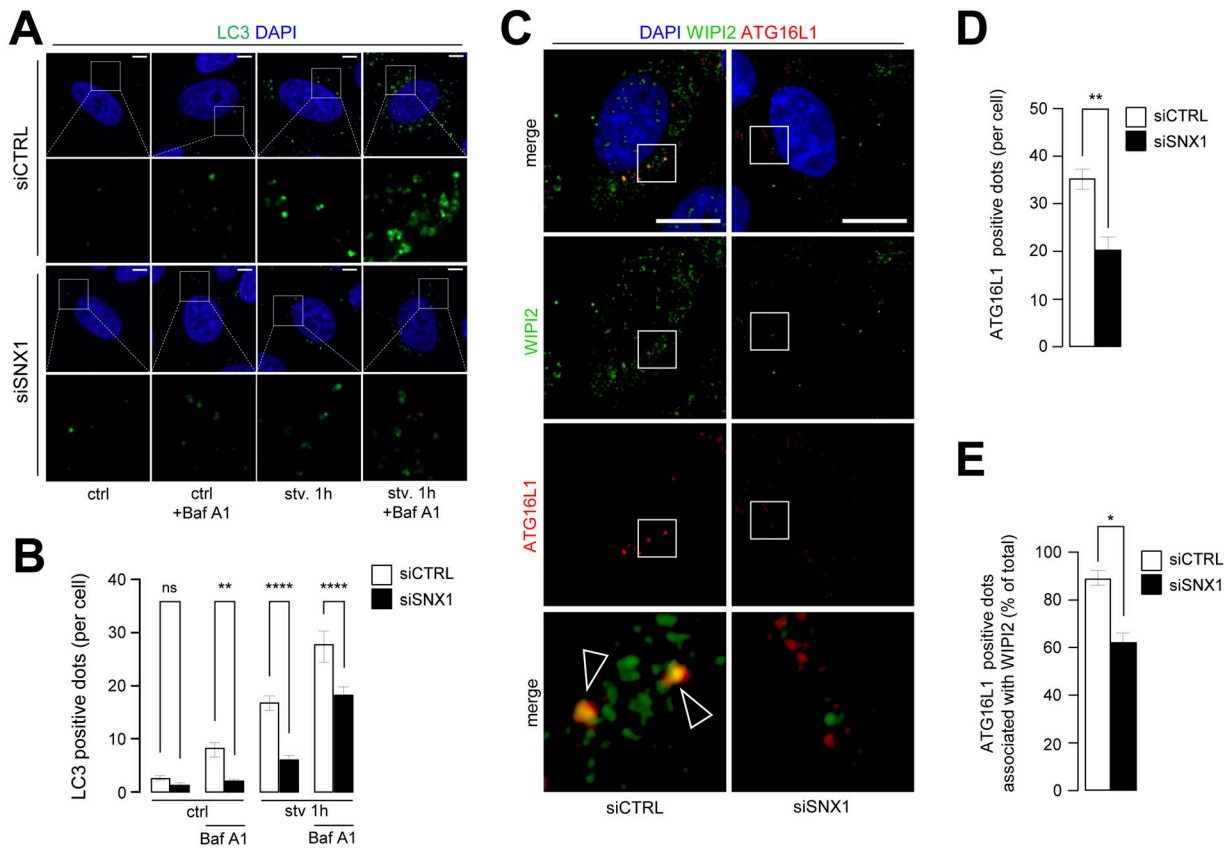

**Figure 3. Knockdown of SNX1 affects autophagic response.**
**(A)** HeLa cells transfected with SNX1 siRNAs (siSNX1) or control siRNAs (siCTRL) were grown under control and starvation conditions (stv. 1 h) with or without 1 h of bafilomycin A1 treatment, fixed and stained with anti-LC3 antibody (green channel) and DAPI. Scale bars = 5 μm. **(A, B)** Quantification of LC3-positive dots per cell from experiment described in (A). Means ± SEM, from three independent experiments; ns, not significant; **P < 0.01, and ****P < 0.0001 in ordinary one-way ANOVA test. **(C)** HeLa cells transfected with SNX1 siRNAs (siSNX1) or control siRNAs (siCTRL) were grown under starvation conditions (1 h), fixed and stained with anti-WIPI2 antibody (green channel), anti-ATG16L1 antibody (red channel), and DAPI. Arrowheads indicate site of ATG16L1 dots associated with WIPI2. Scale bars = 10 μm. **(D)** Quantification of the number of ATG16L1-positive dots per cell. Means ± SEM, from three independent experiments, **P < 0.01 in unpaired two-tailed t test. **(E)** Quantification of the percentage of ATG16L1 dots associated with WIPI2 signal over the total number of ATG16L1 dots. Means ± SEM, from three independent experiments; *P < 0.05 in unpaired two-tailed t test.

key proteins of the phagophore-associated machinery (Melia et al, 2020). As shown in Fig 3C, the total number of ATG16L1 puncta is diminished in siSNX1 cells (Fig 3D) and more importantly, the colocalization of ATG16L1 and WIPI2 is significantly decreased in cells knocked down for SNX1 (Fig 3E). Conversely, overexpression of SNX1 (using increasing cDNA amounts) leads to an artificial boost of the autophagic response, as indicated by the increase of both LC3.II levels (Fig S4A and B) and LC3-positive structures (Fig S4C and D), in normal and starved conditions, confirming the role of SNX1 in the autophagic processes.

## SNX1 endosomal tubules connect ER subdomain fostering autophagosome biogenesis

To investigate further the relationship between starvation-induced mobilization of early endosomes and autophagic response, we tested whether SNX1-associated endosomal membrane specialization upon autophagy induction relates to autophagosome biogenesis sequence. We live-imaged cells transiently transfected with SNX1-GFP and DCFP1-RFP, a key marker of autophagosome

biogenesis associated with PI3P-positive ER membranes (Axe et al, 2008). Live imaging time-lapse acquisitions after 15 min of starvation show that SNX1-GFP tubules transiently connect with DFCP1-RFP structures in a kiss-and-run fashion (Fig 4A), suggesting that SNX1 endosomal tubules can traffic to autophagosome-generating ER membranes. Further analysis was carried out by processing Z-stacks confocal acquisitions of cells transfected with Sec61β-RFP (marker for ER), DFCP1-GFP, and stained with anti-SNX1 antibody. To specifically quantify the proportion of SNX1 structures in close vicinity with DFCP1 puncta present at the ER surface (i.e., to detect pre-autophagosome events only) confocal acquisitions were treated after a sequential computational workflow of image analyses (using ImageJ, ilastik, and Imaris software and depicted in details in Fig S5). This machine learning based workflow allowed for precise 3D rendering and classification of each SNX1 structures based on their proximity with the ER and DFCP1, resulting in a percentage of SNX1 structures recruited to pre-autophagosome events. Data obtained from cells subjected to short time of starvation and analyzed through this workflow reveal that the codistribution of SNX1-positive membranes (seen as vesicles or very

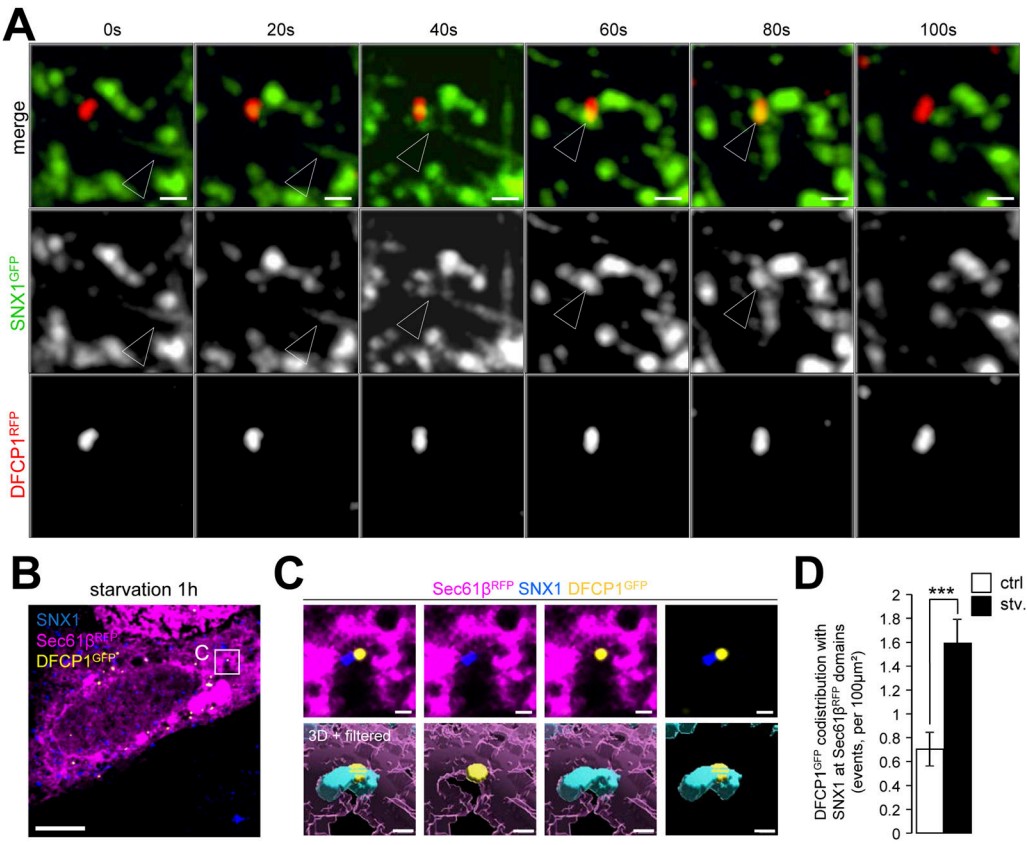

**Figure 4. A pool of SNX1-positive membranes at vicinity of nascent autophagosomes.**
**(A)** Time-lapse images of HeLa cells expressing SNX1-GFP and DCFP1-RFP under starved conditions (stv., 15 min). Arrowheads indicate SNX1-GFP–positive tubules associated with DFPC1-RFP structures. Scale bars = 1 μm. **(B)** HeLa cells expressing Sec61β-RFP (magenta) and DFCP1-GFP (yellow) were grown under starvation for 1 h, fixed and stained with anti-SNX1 (blue) antibodies and observed by confocal microscopy. Scale bar = 5 μm. **(B, C)** Magnified area (×4.5) from confocal image (B), top panel, showing codistribution event between Sec61β-RFP (magenta), SNX1 (blue), and DFCP1-GFP (yellow) and 3D rendering view of magnified area, bottom panel, showing the surface of Sec61β-RFP (magenta), SNX1 (blue), and DFCP1-GFP (yellow). Scale bars = 1 μm. **(C, D)** Quantification of SNX1 codistribution with DFCP1-GFP and Sec61β-RFP (ER)-positive domains per 100 μm² as illustrated in (C). Means ± SEM, from three independent experiments; ***P < 0.001 in unpaired two-tailed t test.

short tubules because of the use of anti-SNX1 antibodies) with the ER-associated omegasome marker DFCP1 increases with starvation time (Fig 4B–D). Similar increase in codistribution is observed between SNX1-positive membranes and ER-associated LC3 by live imaging (Fig S6A) and by 3D acquisitions analyses (Fig S6B–D), although we did not detected any alteration of SNX1 codistribution with the late endosome/lysosome marker LAMP2A (Fig S6E–G), showing that such endosomal membrane and pre-autophagosomal membrane encounter was restricted to early endosomes positive for SNX1. These results show that endosomal SNX1-positive structures transiently appear in the immediate vicinity of ER-membrane sites decorated with pre-autophagic markers in the early stages of the autophagic response.

## Starvation-induced SNX1 endosomal tubules are positive for SNX2 and tether ER membrane via VAPB

SNX proteins have been reported to act as monomers and dimers (Gallon & Cullen, 2015). SNX1 can partner with SNX2, another PI3P-binding and BAR domain protein, notably in retromer-associated endosomal trafficking (Johannes & Wunder, 2011; Gallon & Cullen,

2015). We co-stained fixed cells with anti-SNX1 and anti-SNX2 antibodies and observed an alteration of their mutual codistribution upon starvation (Fig 5A and B). We then co-transfected cells with SNX1-RFP and SNX2-YFP to better monitor endosomal tubule dynamics, and we show that whereas SNX1-RFP tubules are homogeneously positive for SNX2-YFP in control conditions (Fig 5C, top panel), the elongated SNX1 tubules which appear upon very short times of starvation (see also Figs 2 and S2) display a heterogeneous punctate pattern of SNX2-YFP (Fig 5C and D). These results suggest that whereas still being associated to SNX1 endosomes, SNX2 redistributes along SNX1-positive tubules in response to autophagy induction. To go further and to link this observation with data obtained on pre-autophagic subdomains of the ER (Fig 4), we analyzed the relative codistribution of SNX1/SNX2–positive tubules with the autophagic marker LC3. Quantifications based on fluorescence intensity along SNX1-GFP–positive tubules (Fig 5E) show that the maximal intensity of SNX2 staining along the starvation-induced SNX1 tubule correlates with LC3 staining (Fig 5F and G). Similar results were obtained with ATG16L1 (Fig S7A–C). Thus, upon starvation, SNX2 localizes in a punctate pattern with autophagic structures along SNX1 tubules.

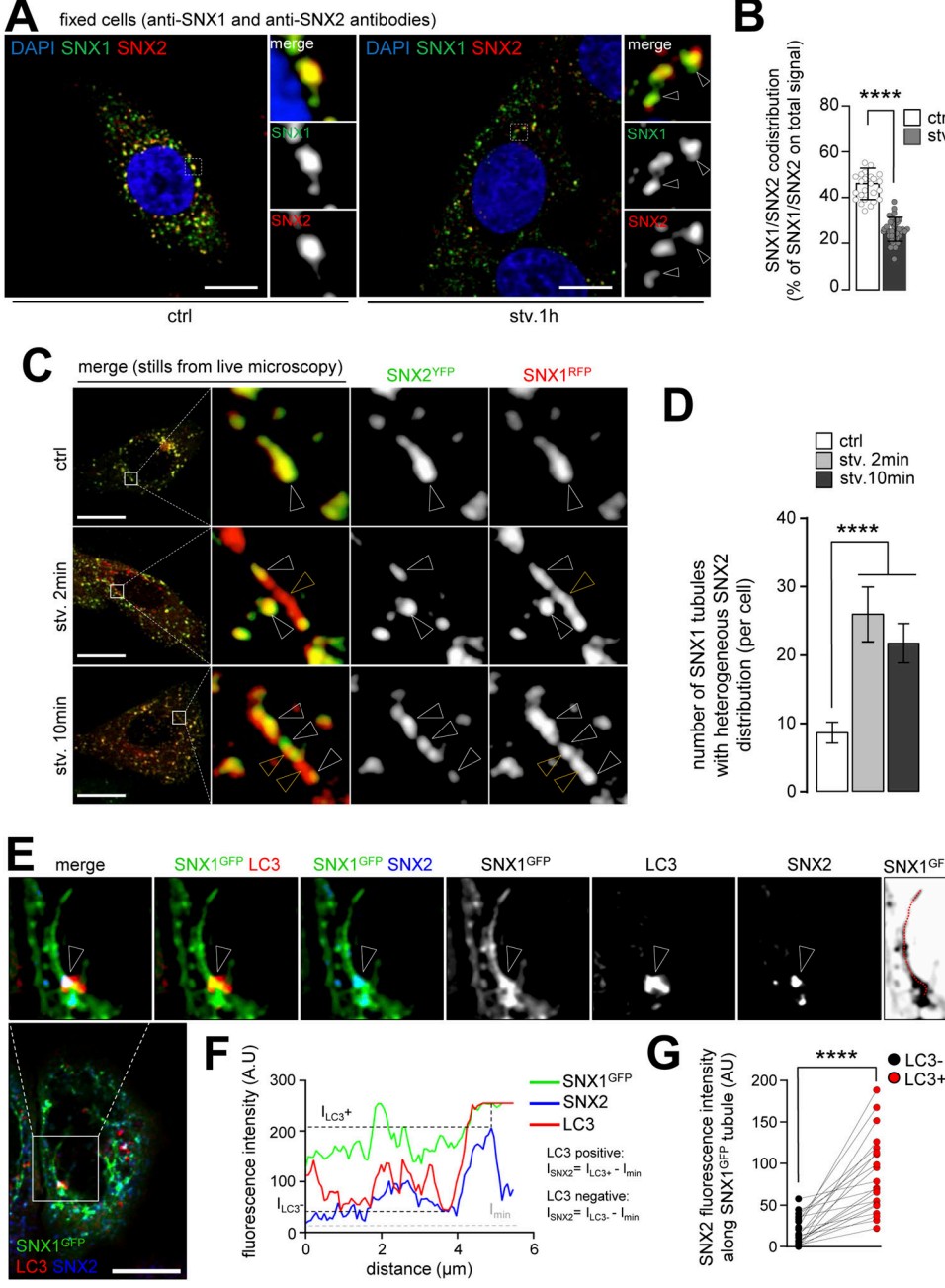

**Figure 5. Association of SNX1 tubules with LC3 is accompanied by a local enrichment of SNX2.**

**(A)** HeLa cells were grown under control (ctrl) or starvation for 1 h (stv. 1 h) fixed and stained with anti-SNX1 (green channel) and anti-SNX2 (red channel) antibodies. Magnified areas show vesicular morphology associated with SNX1 and SNX2, white arrows show changes in SNX1 and SNX2 morphology. Scale bars = 10 μm. **(A, B)** Quantification SNX1 and SNX2 merging fluorescence percentage over total fluorescence of both channels, in conditions depicted in (A). Means ± SD, from three independent experiments; ****$P < 0.0001$ in unpaired two-tailed $t$ test. **(C)** Stills from live microscopy of HeLa cells transfected with SNX2-YFP and SNX1-RFP and grown under control condition or under starvation for short times (2 and 10 min). White arrowheads show colocalization of SNX1 and SNX2 signal, yellow arrowheads show lack of colocalization. Scale bars= 10 μm. **(C, D)** Quantification of SNX1-GFP tubules number displaying heterogenous SNX2-YFP pattern (percentage reported over cell area), in conditions depicted in (C). Means ± SD, from three independent experiments; ****$P < 0.0001$ in ordinary one-way ANOVA. **(E)** HeLa cells were transfected with SNX1-GFP, grown under starvation for 1 h, fixed and stained with anti-SNX2 (blue channel) and anti-LC3 (red channel) antibodies. Top panel shows magnification of a representative local codistribution of LC3 and SNX2 on SNX1-GFP tubule. Arrowheads point site of triple codistribution. Scale bar = 10 μm. **(E, F)** Representative RGB (red-green-blue) profile plot showing the fluorescence intensity of each channel along the SNX1-GFP tubule showed in ((E), right panel). **(G)** Written formulas are the one used for (G) quantifications: the fluorescence intensity of SNX2 at X coordinate where LC3 show the highest (LC3+) or the lowest (LC3−) fluorescence intensity subtracted by the minimal SNX2 fluorescence intensity of each RGB plot. **(F, G)** Values of SNX2 fluorescence intensity along SNX1-GFP tubules depending on the enrichment of LC3 at this spot (LC3+ of LC3−), quantified using the formulas written in (F). Each connected dots correspond to one SNX1-GFP tubule, quantification from three independent experiments, ****$P < 0.0001$ in two-tailed paired $t$ test.

Our data suggest that upon early response to starvation, SNX2 becomes enriched on subdomains of the SNX1 tubules that connect with areas of the ER carrying the early autophagic machinery-associated ATG16L1 protein. Endosomal SNX2 was reported to interact in trans with the ER-associated VAPB protein, leading to tethering zones between endosomes and the ER membrane (Dong et al, 2016). This SNX2/VAPB interaction is required for local endosomal actin nucleation and PI4P transfer to the ER membrane (Dong et al, 2016). Because VAPB is present at the ER autophagosome biogenesis site with key proteins involved in phagophore assembly such as ULK1, FIP200, and WIPI2 (Zhao et al, 2018), we studied whether SNX1/SNX2 endosomal relocalization upon

starvation involved tethering at the ER subdomains through VAPB. Using 3D expansion microscopy (detailed in Fig S8 and illustrated in Video 1) we show that in starved conditions, a subset of SNX1-positive tubules are specifically associated to VAPB-enriched regions of the ER (Fig 6A). To go further, we quantified the local SNX2 intensity along the SNX1 tubules after starvation in regards of VAPB signal: as shown in Fig 6B and C, SNX2 puncta from SNX1 tubules are systematically enriched when codistributing with VAPB. Importantly, using the same experimental frame, we show that VAPB presence in the immediate vicinity of SNX1 tubules significantly increased area positive for early autophagic markers such as WIPI2 (Fig 6D and E) and ATG16L1 (Fig 6F and G). These results suggest that

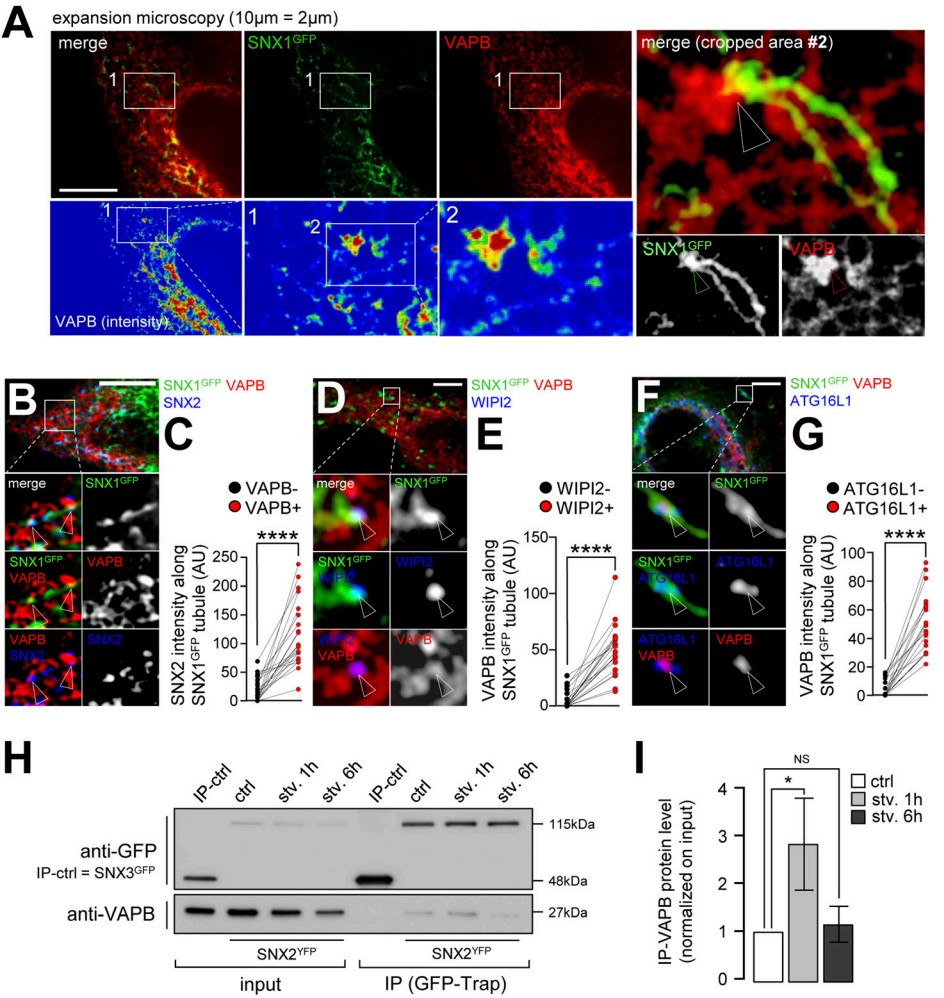

**Figure 6. SNX2–VAPB interaction mediates the tethering of SNX1 tubules to the ER membrane.**
**(A)** Expansion microscopy acquisition of HeLa cells transfected with SNX1-GFP and VAPB-mCherry, grown under starvation for 1 h and fixed. Top panel show confocal microscope acquisition after expansion process (with a scale bar of 10 μm corresponding to 2 μm before expansion). Bottom panel show rainbow colors representing VAPB fluorescence intensity from lowest (blue) to highest (red). Right panel show magnification (crop 2) of a representative SNX1-GFP tubule tethered to VAPB-enriched area. Arrowheads point the site of tethering. **(B)** HeLa cells were transfected with SNX1-GFP, grown under starvation for 1 h, fixed and stained with anti-SNX2 (blue channel) and anti-VAPB (red channel) antibodies. Arrowheads show sites of triple codistribution. Scale bar = 10 μm. **(C)** Quantification of SNX2 fluorescence intensity along the SNX1-GFP tubules depending on the enrichment of VAPB at this spot (VAPB+ in red or VAPB– in black) using the same method as Fig 5F. Each connected dots correspond to one SNX1-GFP tubule, quantification from three independent experiments, ****P < 0.0001 in two-tailed paired t test. **(D)** HeLa cells were transfected with SNX1-GFP, grown under starvation for 1 h, fixed and stained with anti-WIPI2 (blue channel) and anti-VAPB (red channel) antibodies. Arrowheads show sites of triple codistribution. Scale bar = 5 μm. **(E)** Quantification of VAPB fluorescence intensity along the SNX1-GFP tubules depending on the enrichment of WIPI2 at this spot (WIPI2+ in red or WIPI2– in black) using the same method as Fig 5F. Each connected dots correspond to one SNX1-GFP tubule, quantification from three independent experiments, ****P < 0.0001 in two-tailed paired t test. **(F)** HeLa cells were transfected with SNX1-GFP, grown under starvation for 1 h, fixed and stained with anti-ATG16L1 (blue channel) and anti-VAPB (red channel) antibodies. Arrowheads show sites of triple

codistribution. Scale bar = 5 μm. **(G)** Quantification of VAPB fluorescence intensity along the SNX1-GFP tubules depending on the enrichment of ATG16L1 at this spot (ATG16L1+ in red or ATG16L1– in black) using the same method as Fig 5F. Each connected dots correspond to one SNX1-GFP tubule, quantification from three independent experiments, ****P < 0.0001 in two-tailed paired t test. **(H)** Western blots of total cell lysate (input) and immunoprecipitates (IP) of HeLa cells transfected with SNX2-YFP or SNX3-GFP (IP-ctrl); cells were grown in complete medium or starved for 1 or 6 h (stv. 1 h, stv. 6 h). Samples were immunoprecipitated using GFP-trap beads (GFP and YFP) and immunostained with VAPB or GFP antibodies. **(F, I)** Quantification of immunoprecipitated VAPB protein level over input level, in conditions depicted in (F); means ± SD, from five independent experiments; *P < 0.05 in ordinary one-way ANOVA test.

SNX1-positive endosomal membranes generated in response to starvation are stabilized in the vicinity of VAPB-associated regions of the ER which carry autophagic machinery.

To decipher whether SNX2 ability to bind VAPB is modified by autophagy-inducing conditions, we performed GFP-trap–based co-immunoprecipitations in cells transfected with SNX2-YFP and starved for 1 and 6 h, using an endosomal SNX lacking the predictive domain for VAPB binding (SNX3 [Pons et al, 2008]), as negative control. We observed that the interaction between SNX2-YFP and endogenous VAPB significantly increased after 1 h of starvation and was back to basal levels after 6 h of starvation (Fig 6H and I). These results reveal that after a short period of starvation, in the timeframe of autophagosome biogenesis that we report to be associated with massive SNX1 endosomal tubulation (Fig 2), the interaction between endosomal SNX2 and VAPB is transiently stabilized, thus suggesting that SNX2–VAPB tethering is mobilized during early stages of autophagosomal processes.

To test whether the SNX–VAPB complex is required for the endosomal SNX1 mobilization associated with autophagosome biogenesis associated membrane subdomains, we analyzed the effects of SNX2 knockdown on SNX1-associated tubulation in response to starvation. In cells transfected with siSNX2 (Fig 7A and B), endosomal tubulation associated with SNX1-GFP no longer responded to starvation (Fig 7C and D) when comparing to siCTRL cells. In this context, we report that the knockdown of SNX2 does not alter SNX1 levels (and vice versa, Fig S9), supporting the hypothesis that SNX1 and SNX2, although often directly associated, could have independent biological functions. To decipher the role of SNX2 in this process, we questioned its putative regulatory function in the stabilization of SNX1-positive endosomal tubules with autophagy early stages associated ER sub-domains (Figs 4–6) in response to starvation. Because we hypothesize that the transient membrane tethering between SNX1 endosomes and ER subdomains could occur via a SNX2 to VAPB trans-interaction, as previously reported (Dong et al, 2016), we tested

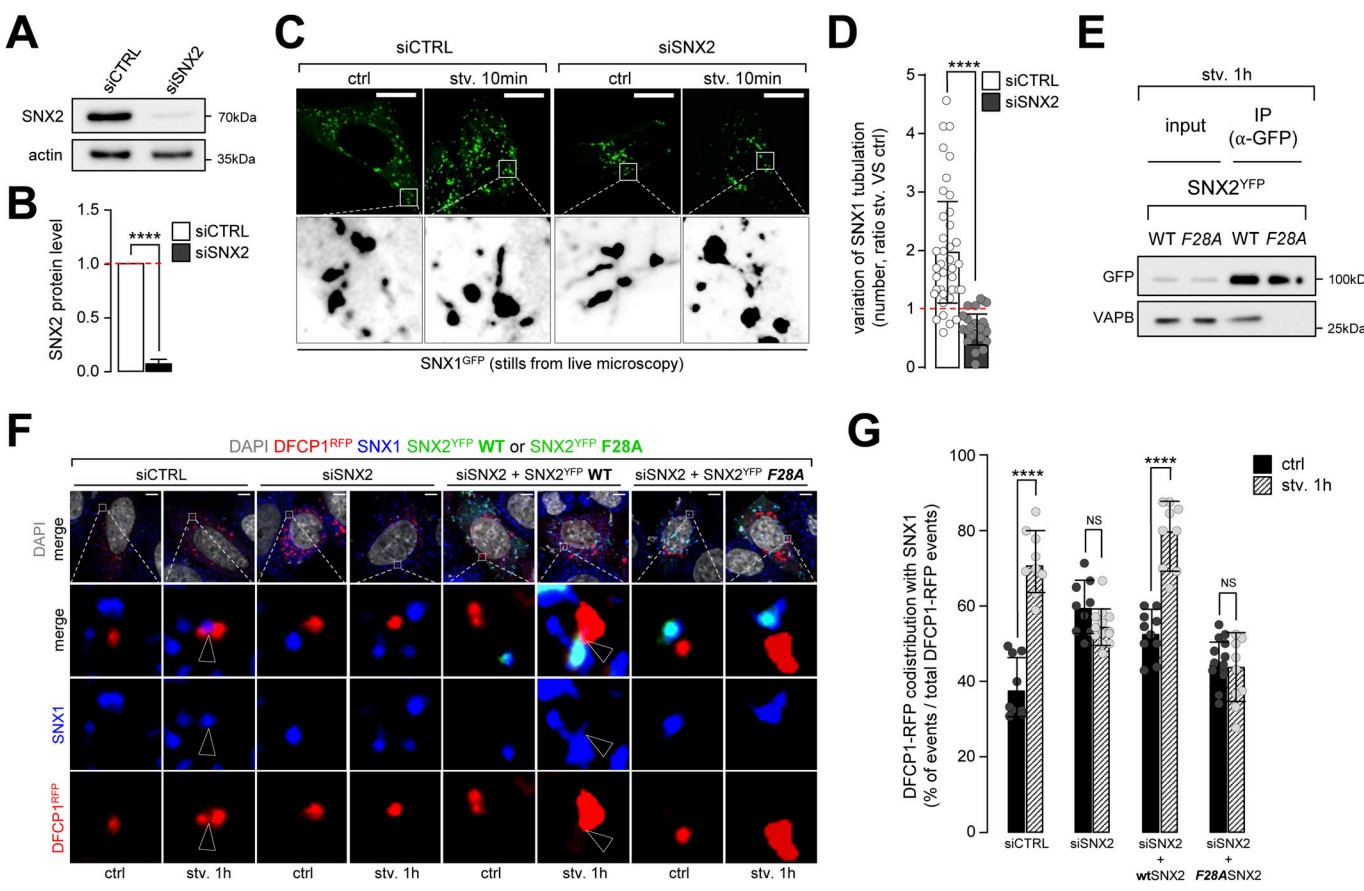

**Figure 7. Recruitment of SNX1-positive membranes to autophagosome biogenesis site depends on SNX2.**
**(A)** HeLa cells were transfected with SNX2 siRNAs (siSNX2) or control siRNAs (siCTRL). Representative Western blot of corresponding lysates for SNX2 are shown. Actin was used as a loading control. **(A, B)** Quantification of SNX2 protein levels from Western blots as shown in (A). Means ± SD, from three independent experiments; ****$P < 0.0001$ in unpaired two-tailed $t$ test. **(C)** Stills from live microscopy of HeLa cells transiently expressing SNX1-GFP grown in complete medium (ctrl) or under starvation for 10 min (stv. 10 min) transfected with siRNA for SNX2 (siSNX2) or control siRNA (siCTRL). Enlarged crops show SNX1-GFP–positive tubules. Scale bars = 10 $\mu$m. **(D)** Quantification of SNX1-GFP tubules forming in cells starved for 10 min (stv) normalized on the number of tubules in complete medium (ctrl) in siCTRL or siSNX2 transfected cells; means ± SD, from three independent experiments; ****$P < 0.0001$ in unpaired two-tailed $t$ test. **(E)** Western blots of total cell lysate (input) and immunoprecipitates (IP) of HeLa cells transfected with SNX2-YFP WT or SNX2-YFP F28A, grown in starved condition for 1 h (stv. 1 h). Samples were immunoprecipitated using GFP-trap beads (GFP and YFP) and immunostained with VAPB or GFP antibodies. **(F)** HeLa cells transfected with either siCTRL or siSNX2 alone or rescued with SNX2-YFP WT or SNX2-YFP F28A and transiently expressing DFCP1-RFP were grown in complete medium or under starvation for 1 h, fixed and stained with anti-SNX1 (blue) antibody. Arrowhead points site of codistribution of SNX1 with DFCP1. Scale bar = 5 $\mu$m. **(F, G)** Quantification of the percentage of DFCP1-RFP puncta codistributing with SNX1 over the total number of DFCP1-RFP puncta per cell, in conditions depicted in (F); means ± SD, from three independent experiments; ****$P < 0.0001$ in ordinary two-way ANOVA.

the effect of a SNX2 mutant, in which the VAPB interacting domain is altered (F28A-SNX2, Fig 7E). We show that upon starvation, SNX1 failed to co-distribute with the ER-associated DFCP1 pre-autophagic marker in siSNX2 cells (Fig 7F and G, siCTRL and siSNX2 conditions). Importantly, overexpression of wild type (wtSNX2) rescues the phenotype induced by starvation, the overexpression of the mutated form of SNX2 failed to do so (Fig 7F and G, siSNX2 + wtSNX2 and siSNX2 + F28A-SNX2 conditions), suggesting that SNX2–VAPB binding is required to target SNX1-positive endosomal membranes to the sites of autophagosome biogenesis at ER membrane.

## Discussion

Cell starvation induces nutritional stress, triggering a cascade of adjustments in both physiological and pathological backgrounds.

Deprivation of nutriments such as amino acids induces numerous signaling pathways such as the mTOR cascade. It also induces the biogenesis of autophagosomes in the vicinity of the ER through membrane contact sites mobilization (Morel, 2020). However, how it affects the overall cellular endomembranes is not yet clearly documented. Recent data point out the requirement for early endosomal equipment in the primary stages of autophagosome assembly. For instance, recycling endosomes contribute to deliver ATG proteins to autophagosome biogenesis sites (Longatti & Tooze, 2012; Puri et al, 2013, 2018; Dooley et al, 2014; Wilson et al, 2014; Judith et al, 2019). Despite recent studies describing lysosomal membrane tubulation and recycling in response to stress (Rong et al, 2012; Li et al, 2016; Zhao et al, 2020; Zhou et al, 2022), the early consequences of starvation on the overall endosomal system—and its putative connection to autophagosome biogenesis sequence—are still poorly understood.

We hypothesized that delivery of specific early endosomal membranes to the ER subdomains engaged in autophagic processes could participate in the local regulation of membrane dynamics responsible for omegasome/phagophore assembly in response to starvation.

We explored the role of the membrane-shaping SNXs family, because on one hand, some of its members participate in stress response and autophagic processes (Knævelsrud et al, 2013; Søreng et al, 2018; Antón et al, 2020; Dong et al, 2020; Ravussin et al, 2021), and on the other hand, they are characterized as early endosomal membrane modulators, promoting vesicular or tubular extensions in response to certain stimuli (Johannes & Wunder, 2011; Gallon & Cullen, 2015).We report that upon short starvation known to induce bona fide autophagosome biogenesis, a rapid and transient alteration of the early endosomes morphology takes place, driven by a local mobilization of SNX1. SNX1 is required for autophagosome biogenesis because its depletion impairs the autophagic flux. Starvation-induced SNX1 endosomal tubules are transiently connected to the ER membrane, on pre-autophagic machinery assembly sites. The endosomal SNX1 tubules–ER subdomains tethering is controlled by SNX2, reported to interact with SNX1 in its retromer function (Burd & Cullen, 2014; Gallon & Cullen, 2015), and VAPB, a partner of SNX2 which promotes trans-interaction between endosomes and ER membranes (Dong et al, 2016). Initially homogenously colocalizing with SNX1, starvation induces a local modification of the SNX2 patterning on the SNX1 tubules emerging from early endosomes. Indeed, SNX2 forms patches strongly codistributing with VAPB-positive areas on the ER domains, fostering autophagosome assembly, as assessed by the presence of WIPI2 and ATG16L1. Interaction between SNX2 and VAPB, occurring via the F28 residue on SNX2, is required for SNX1 endosomal tubule recruitment to the ER sites of autophagosome biogenesis, where their tethering to VAPB is transiently stabilized in the early stages of the starvation response.

Altogether, our data support dynamic connections between SNX1-positive endosomal tubules and ER sites of autophagosome biogenesis. It is thus tempting to hypothesize that early endosomal membranes participate in the onset of autophagy after cellular starvation, with a mechanism involving SNX1 on the early endosomes, SNX2 as a bridge, and VAPB as an ER tether.

Phosphoinositides are master organizers of membrane compartment identity (Posor et al, 2022). On the ER membrane, PI3P is a central player in starvation-associated autophagy (Claude-Taupin & Morel, 2021). Early endosomes are also natural pools of PI3P and PI3P-binding proteins. Whether the endosomal PI3P associated with SNX1 is directly required for the pre-autophagic steps at the ER membrane will have to be investigated to further decipher the dialog between endosomal and non-endosomal membrane during the biogenesis of autophagosomes. In this context, our data argue for the need of an endosome-to-omegasome local—and transient—PI3P transfer investigation, which could explain at some point why both endosomes and ER membranes have been proposed to sustain autophagosome biogenesis. Moreover, the requirement of PI4P-associated SNX2/VAPB physical crosstalk (Dong et al, 2016) upon endosome-ER tethering during early starvation questions a putative PI4P transfer from early endosomes to the ER membrane, because PI4P has been shown to be an important

regulator of the early stages of autophagosome assembly (Judith et al, 2019).

Interestingly, several SNXs have been reported to participate in stress response. The PI4,5P2-binding SNX18 has been associated with autophagy. It decorates a subset of Rab11 recycling endosomes and is essential in the early steps of autophagosome biogenesis by providing pools of ATG16L1 via membrane tubules (Knævelsrud et al, 2013; Søreng et al, 2018). On the endosome, SNX1 forms heterodimeric combinations with SNX2, SNX6, and SNX5 in the retromer complex that transport cargoes from early endosomes to the TGN (Johannes & Wunder, 2011; Lu & Hong, 2014; Scott et al, 2014). Our data report and describe for the first time how endosomal tubules can be mobilized and tethered to the omegasome area of the ER in response to stress. Addressing whether and how an orchestrated and synchronized implication of various SNXs, binding different phosphoinositides, might participate in the regulation of the early stages of autophagy is essential to further understand an overview of endosomal scaffolding protein mobilization by stress-related membrane dynamics.

The prompt and yet transient SNX1-dependent tubulation of early endosomes in response to starvation could illustrate a more general endomembrane mobilization to face the consequences of stress in a concerted manner. The origin of such rapid membrane remodeling of SNX1 endosomal compartment is to be characterized in the future, notably by questioning the putative implication of endosomal small GTPases Rab5 and Rab11 in these processes. During starvation, cytoplasm alkalization was suggested to participate in lysosome redistribution to the perinuclear zone (Korolchuk et al, 2011) and recent data report that a SNX19-associated membrane tethering with the ER regulates late endosomes/lysosomes mobility (Saric et al, 2021). Thus, organelles spatial distribution and their ability to alter their morphology may be hallmarks of an early response to stress to secure the optimal conditions promoting the initiation of autophagy. Our results open up a new path to investigate how mammalian cells coordinate several endomembranes mobilization and intracellular compartments to face and adapt to starvation.

# Materials and Methods

### Cell culture, cell treatments, and plasmids

HeLa cells from ATCC were grown in MEM, supplemented with GlutaMAX (41090093; Thermo Fisher Scientific), 10% FCS, and non-essential amino acid mix (11140035; Gibco, Thermo Fisher Scientific) at 37°C and 5% $CO_2$. MEF cells from ATCC were grown in DMEM (41966; Gibco) supplemented with 20% FCS at 37°C and 5% $CO_2$. For starvation analyses, cells were washed twice with 1× PBS and incubated in Earle's Balanced Salt Solution (EBSS, 24010043; Thermo Fisher Scientific) for the indicated times. Rapamycin (R0395; Sigma-Aldrich) was added at 200 nM final concentration for the indicated times. Autophagic flux was monitored using the lysosomal inhibitor Baf A1 (B1793; Sigma-Aldrich) at 100 nM final.

Transfections of cDNA plasmids were performed at least 24 h before the experiments using FuGENE (E2312; Promega) according to manufacturer's recommendations. Regular transfections were

made at 5 ng/μl, whereas overexpression was performed with 2 or 4 μg/μl of cDNA. DFCP1-RFP and DFCP1-GFP were from Addgene (58765). Sec61β-RFP cDNA was from T Rapoport, SNX1-GFP cDNA was from GN Gill, SNX3-GFP cDNA was from V Pons and J Gruenberg, and SNX2-YFP and VAPB-mCherry cDNAs were from P De Camilli. SNX2-YFP F28A was made using the QuikChange Multi Site-Directed Mutagenesis Kit (200515-5; Agilent Technologies) (primers: FW: 5'-gacggagaggacctggccaccagcactgtctc-3' RV: 5'-gagacagtgctggtggccaggtcctctccgtc-3'). SNX1-RFP was made using NEBuilder HiFi DNA Assembly Master Mix (E2621L; New England BioLabs) from Sec61β-RFP and SNX1-GFP. LC3-RFP was from Addgene (21075).

Knockdown of human SNX1 and SNX2 was performed using a pool of siRNAs (SI00047775 and SI00047782 for SNX1, SI04190907 and SI04206475 for SNX2; QIAGEN) transfected at 20 nM with Lipofectamine RNAiMax (13778150; Thermo Fisher Scientific) 48-h prior experiment.

## Antibodies

The EEA1 protein was labeled by anti-EEA1 mouse antibodies from BD TransLab (610456) for both immunofluorescence (IF, dilution 1/200). Actin protein was used as a loading control for WB with anti-Actin rabbit antibodies from Sigma (A2066, dilution: 1/10,000). The LC3B protein was labeled by anti-LC3B antibodies from MBL, either mouse (152-3, dilution 1/200) or rabbit (PM036, dilution 1/150) for IF, and rabbit antibodies from Sigma-Aldrich (L7543, dilution 1/3,000) were used for WB detection. SNX1 protein was detected by anti-SNX1 rabbit antibodies from Novus Biologicals (NBP2-56957) for both IF (dilution: 1/150) and WB (dilution: 1/1,000). SNX2 protein was detected using anti-SNX2 mouse antibodies from BD Trans Laboratories (611308) for both IF (dilution: 1/150) and WB (dilution: 1/3,000). VAPB protein was targeted using anti-VAPB rabbit antibodies from ProteinTech (14477-1-AP) for IF (dilution: 1/150) and mouse antibodies from ProteinTech (66191-1-Ig) for WB (dilution 1/3,000). GFP was targeted by mouse anti-GFP in WB (sc-9996, dilution: 1/3,000). WIPI2 protein was detected using anti-WIPI2 mouse antibody for IF (ab105459, dilution 1/150). ATG16L1 protein was labeled in IF by rabbit anti-ATG16L1 antibody (PM040, dilution: 1/150). LAMP2A protein was detected in IF by mouse anti-LAMP2A (sc-18822, dilution: 1/150).

## SDS–PAGE and Western blotting

For biochemical analyses, cells were washed twice with ice-cold PBS and lysed using Laemmli1X (60 mM Tris–HCl pH = 6.8, 2% SDS, 10% glycerol, bromophenol blue, supplemented with 100 mM DTT) for 10 min. Samples were boiled for 15 min at 95°C, separated by 13.5% acrylamide SDS–PAGE, and transferred onto PVDF membranes. Western blot analysis was performed with specific primary antibodies diluted in PBS-Tween 0.1%–BSA 5% overnight at 4°C. Secondary HRP-conjugated anti-rabbit IgG (AP308P; Chemicon) and HRP-conjugated anti-mouse IgG (1706516; Bio-Rad) were diluted in PBS-Tween 0.1% and milk 5% and incubated for 2 h at room temperature with PVDF membranes. Bands were visualized using a chemiluminescent HRP substrate (WBKLS0500; MERCK) and a ChemiDoc MP Imaging System (Bio-Rad). Quantification of band intensities was carried out using Fiji software, and protein levels are normalized on actin.

## Immunofluorescence and analysis

For classical immunofluorescence analyses, cells were seeded onto 12-mm glass coverslips and fixed with 4% PFA pre-warmed at 37°C for 25 min. Cells were then washed with PBS and blocked with 10% FCS in PBS for 30 min. Incubation with primary antibodies diluted in permeabilization/blocking buffer (PBS with 10% FCS and 0.05% saponin) was done for 2 h at room temperature. Coverslips were PBS-washed before and after adding secondary antibodies for 1 h in blocking buffer. Coverslips were mounted on microscope slides using Mowiol containing DAPI. For LC3 puncta counting experiments, images were obtained using a Zeiss ApoTome.2 fluorescence microscope. All other immunofluorescence experiments were observed using either Zeiss LSM 700 confocal microscope or Zeiss Spinning Disk, both using a 63× oil-immersion objective. Image analysis was performed using Zen Light Zeiss LSM software, in which the background was reduced using brightness and contrast adjustments applied to the whole image. EEA1-positive endosomes morphological analysis was made using Icy software and analyses of ROIs with roundness (normalized ratio between the radius of the minimum inscribed and largest circumscribed circle, expressed as a percentage) and elongation (scale factor given by the ratio between the first and the second ellipse diameters where a non-elongated object has a minimal value of 1) parameters. The analysis of codistribution of SNX1 and EEA1 was addressed by selecting specific vesicular signal using machine learning (ilastik1.3.3post3), then convert it into binary mask and quantify merging area using color threshold with Fiji software. LC3 and ATG16L1 puncta quantifications were made using Spot Detector plugin in Icy software. Correlation of fluorescence intensity of two markers along each SNX1-GFP tubule using RGB profile plots was made with Fiji software and Color profiler plugin (Color_Profiler. jar, https://imagej.nih.gov/ij/plugins/color-profiler.html). A line was drawn on each of the SNX1-GFP tubules to obtain the specific RGB profile plot for all three channels. On each RGB plot (X and Y axes) we looked at the X coordinate where the first channel (LC3 in Fig 5E–G, ATG16L1 in Figs S7A–C and 6F and G, VAPB in Fig 6B and C, WIPI2 in Fig 6D and E) exhibited the highest and the lowest fluorescence intensity according to Y axe. Keeping the same X coordinate, we looked at the Y value of the second channel (SNX2 in Figs 5E–G, S7A–C, and 6B and C, VAPB in Fig 6D–G) and subtracted this value to the minimum intensity of this second channel on each RGB plot to normalize all experimental measurements.

## Live imaging microscopy and tubule analysis

HeLa cells were transfected with indicated cDNAs 36 h prior live recording and seeded in an eight-wells IBIDI cell chamber (80826; IBIDI). Imaging was performed with Zeiss Spinning Disk Microscope using a 63× oil-immersion objective. Tubule's quantification was performed manually using Icy software (http://icy.bioimageanalysis.org) on z-projections. 4D images were taken with 3 μm depth and 10 s interval for 90 s.

### Z-stack confocal acquisitions subjected to 3D rendering and machine learning detection

Z-stack confocal images were obtained with Zeiss LSM 700 confocal microscope with 0.08-$\mu$m interval slices and 0.07 $\mu$m pixel size. Then, channels were split using ImageJ 2.1.0, and each channel was independently subjected to ilastik 1.3.post3 software for machine learning detection. Machine learning was based on 31 features with sigma 0.30-1.60 in 3D and 3.50–5.00 in 2D (Gaussian Smoothing, Laplacian of Gaussian, Gaussian Gradient Magnitude, Difference of Gaussians, Structure Tensor Eigenvalues, and Hessian of Gaussian Eigenvalues). Selection of signal was manually chosen based on vesicular/tubular aspects. Data were exported as simple segmentation in .tiff format and converted to binary mask. Raw and segmented channels were merged using again ImageJ 2.1.0 and then analyzed in Imaris ×64 9.7.1. To allow better 3D visualization, segmented channels were detected as surfaces by Imaris ×64 9.7.1. with no further changes in the settings. Classification of the resulting surface of SNX1 structures was made using "classification" tool of Imaris software based on the shortest distance feature with ER surface and DFCP1, LC3, and LAMP2A surfaces, thus allowing the quantification of the percentage of triple codistribution events over the total of SNX1 structures.

### Expansion microscopy

This protocol is based on the combination of two expansion microscopy protocols found in Chozinski et al (2016) and Gambarotto et al (2019). Fixed and stained coverslips were incubated with 0.1 mg/ml Acryloyl-X (A20770; Invitrogen) at 4°C overnight. Coverslips were quickly (1–2 min) incubated at room temperature with monomer solution (PBS 1×, 19% sodium acrylate, 10% acrylamide, and 0.1% N,N′-methyl-enebisacrylamide). Gelation was performed by putting coverslips with cells facing a 35 $\mu$l drop of monomer solution supplemented with 0.5% APS and 0.5% TEMED on Parafilm and incubated for 30 min at room temperature. Coverslips with gels were transferred into 2 ml of digestion buffer (Tris-acetate-EDTA buffer 1×, 0.5% Triton-x100, and 0.8 M guanidine HCl) supplemented with fresh proteinase K (8 U/ml) and incubated for exactly 30 min at room temperature. Gels were collected and carefully transferred into ~50 ml of deionized water for expansion. Water was changed twice every 30 min and incubated overnight at room temperature in the dark for full expansion.

### Immunoprecipitation analysis

Immunoprecipitation analysis was performed using GFP agarose beads (GFP-Trap; ChromoTek). Two million HeLa cells previously transfected with SNX2-YFP or SNX3-GFP were suspended in lysis buffer (10 mM Tris, pH 7.5, 150 mM NaCl, 0.5% mM EDTA, and 0.5% Nonidet P-40) and protease and phosphatase inhibitor cocktail (78443; Pierce). Cell lysates were centrifuged at 15,000 $g$ for 15 min. The resulting supernatant was incubated with anti-GFP beads for 1 h, while rotating at 4°C. Beads were collected by centrifugation and washed four times with wash buffer (10 mM Tris, pH 7.5, 150 mM NaCl, 0.5% mM EDTA, and 0.05% Nonidet P-40). Then, bead-bounded samples were eluted in 2× Laemmli, boiled for 5 min, loaded onto SDS–PAGE gels, and blotted on PVDF membranes as described in the Western blotting section.

### Statistical analysis and reproducibility

Statistical analyses were performed by using GraphPad Prism 9.1.2 (*$P < 0.05$, **$P < 0.01$, ***$P < 0.001$, and ****$P < 0.0001$). In panels showing Western blotting or immunofluorescence analysis, unless stated otherwise, images are representative of three independent experiments with similar results.

## Supplementary Information

## Acknowledgements

We would like to thank Dr Zeina Chamoun (Institut Curie, Paris, France) for manuscript correction and editing. We thank Dr Pons (Inserm UMR 1048, Toulouse, France) and Pr Gruenberg (Dept. of Biochemistry, University of Geneva, CH) for the SNX3 construct. We also thank Pr Rapoport (Boston), Pr Gill (San Diego), and Pr De Camilli (Yale) for sharing their constructs with us. We are grateful to our colleagues Dr Nicolas Dupont and Dr Patrice Codogno from the Institut Necker Enfants Malades (Inserm U1151, Paris, France) for critical reading of the manuscript and helpful advices. We wish to thank Necker SFR technical platforms, especially Nicolas Goudin at the Necker Bio-image Analysis Center and Meriem Garfa-Traoré and Sébastien Dupichaud at the Cell Imaging Facility. This study was supported in part by ANR (Agence Nationale de la Recherche, grants # ANR-17-CE140030-02, ANR-17-CE13-0015-003, and ANR-18-CE14-0026-02 [France]), INSERM (Institut National de la Santé et de la Recherche Médicale [France]), CNRS (Centre National de la Recherche Scientifique [France]), and Université de Paris-Cité (France). J Da Graça is a recipient of a doctoral fellowship from the French Ministry of Research/Université de Paris.

### Author Contributions

J Da Graça: conceptualization, data curation, software, formal analysis, validation, investigation, visualization, methodology, and writing—original draft, review, and editing.
J Charles: data curation, investigation, and methodology.
M Djebar: investigation and methodology.
K Alvarez-Valadez: formal analysis and methodology.
J Botti: investigation and methodology.
E Morel: conceptualization, data curation, software, formal analysis, supervision, funding acquisition, validation, investigation, visualization, methodology, project administration, and writing—original draft, review, and editing.

### Conflict of Interest Statement

The authors declare that they have no conflict of interest.

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
