## [Reviewer comments · Life Science Alliance]

Life Science Alliance

a SNX1-SNX2-VAPB partnership regulates endosomal membrane rewiring in response to nutritional stress

DA GRACA JULIANE, CHARLES JULIETTE, Morgane Djebbar, Karla Alvarez-Valadez, Joelle Botti, and ETIENNE MOREL
DOI: <https://doi.org/10.26508/lsa.202201652>

Corresponding author(s): ETIENNE MOREL, Université Paris Cité

Review Timeline:

Submission Date:	2022-08-04
Editorial Decision:	2022-08-04
Revision Received:	2022-11-23
Editorial Decision:	2022-12-12
Revision Received:	2022-12-18
Accepted:	2022-12-19

Transaction Report:

Reviewer #1 Review

Comments to the Authors (Required):

This study by De Graca et al. focuses on the possible role of SNX1 (and SNX2) during autophagy. This follows recent studies highlighting how interference with SNXs (e.g., SNX18; SNX4; SNX7) alters the starvation autophagy response, typically through ATG9A trafficking defects. Here the authors firstly address the effects of starvation on endosomal behavior, with tubulation of overexpressed SNX1-GFP observed, and apparent changes in EEA1 morphology. The study then goes on to address this with a series of mainly imaging-based experiments with some fairly limited autophagy assays that are restricted to LC3 puncta/lipidation analysis. Whilst there are some interesting initial findings regarding changes in endosomal behavior and direct or indirect effects on autophagy, the mechanistic side of the study is weak and relies heavily on quite subjective assessment of subtle changes in marker localization. The study is also severely weakened by the use of a single cell-line (HeLa) and single siRNAs with limited rescue experiments. In sum, the work is too preliminary and not convincing enough to be published in this journal.

1. T-tests should not be used for $n > 2$ comparisons.
2. What is "endosomal membrane rewiring towards ER sub-domains"? Abstract
3. Fig. 1A: transient increase in EEA1 and Rab11 needs further validation. Is this transcriptionally controlled? Does this involve decreased proteasomal (or even, although unlikely, autophagy)? Normalized against actin, could it not be possible that actin is transiently decreased? In other words, the choice of normalization could lead to anomalies here.
4. EEA1 morphology experiment (Fig. 1C, D) and accompanying SNX1 analysis in Fig. S1 are weak. What is being analyzed in Fig. 1C? Is this measured by eye or has software been used for an objective readout? Why is the analysis in Fig. S1C, D extended out to 24h which is extremely late for starvation analysis?
5. Tubulation of SNX1-GFP tubules is interesting (Fig. 2), but could be forced by overexpression. In any case, this needs to be validated in a different cell-line(s). The accompanying data in Fig. S2 that attempts to demonstrate a role for SNX1 in early endosomal tubulation are poor - the accompanying images of EEA1 do not show any differences in tubular structures and thus appears to be highly subjective.
6. The autophagy response in siSNX1 cells does appear to be blunted. However, only one siRNA is used and there does not appear to be any attempt to rescue this effect. I would also prefer to see complete immunoblots in place of the montage shown in Fig. 3A.
7. The overexpression effect looks convincing (Fig. S3), but its biological significance will always be questionable. The attempt to confirm the relevance of this through use of the R186A mutant (Fig. S4) throws up more questions than it answers because of unusual effects on total LC3 levels. What do the authors suggest is happening here? Is there a dominant negative effect, and if

so what is the mechanism?

8. I was not convinced by the co-localization analysis between SNX1 and DFCP1 (Fig. 4) - since the distribution of SNX1 (punctate to tubular) is a feature of the starvation response, chance colocalization is likely to increase. A further control is needed. For the SNX1-LC3 colocalization, this appears more likely, but it should be noted that RFP-LC3 will still be fluorescent in the lysosome so we may be looking at late autophagosome/lysosomal colocalization here, which is not consistent with a role for SNX1 during early autophagosome biogenesis.

9. The data in Fig. 5 are interpreted as meaning that SNX1 establishes tubules during starvation along which SNX2 foci can be found, and that these have a tendency to co-distribute with LC3. This is potentially interesting but needs more work to validate e.g., other autophagy markers for early stages (e.g., WIPI2; ATG16L1), higher resolution microscopy including correlative light-electron microscopy and immunoEM.

10. Fig. 6, the VAPB colocalization is unconvincing and the quantitation shown in Fig. 6E is poorly defined and makes no sense to me. The IP experiment in Fig. 6F, G is potentially interesting but needs to be shown using native proteins.

11. In Fig. 7 a single siRNA is used to deplete SNX2 to assess the behavior of overexpressed SNX1-GFP. It looks as though SNX1-GFP is already more tubulated before starvation (Fig. 7C), meaning that the analysis as presented in Fig. 7D misses that effect. Fig. 7E, F appears to have been analyzed by eye-balling apparent co-distribution. If so, this lacks objectivity.

Reviewer #2 Review

Comments to the Authors (Required):

This paper describes a role for SNX1 in the tubulation of early endosomal compartment upon nutrient starvation. The authors propose that the SNX1-positive tubulated endosomes can interact through SNX2 with ER subdomains positive for VAPB associated with phagophore biogenesis. The identification of the endosomal compartment as a membrane reservoir for early autophagy is not novel as membrane contributions have been shown to occur from recycling endosomes via Snx18 (Knævelsrud et al. JCB 2013). The authors use a combination of microscopy with image quantification and analysis to support their model and use biochemistry to assess autophagic flux and protein interactions. However, despite the identification of the role of SNX1-SNX2-VAPB as main actors the molecular details of the interaction of the three proteins is not clear. Nor is sufficient detail provided on how each component influences the others in the assembly of this protein/membrane complex upon starvation. Even if the imaging in this paper is of high quality, the methodology used for the analysis of the data is many times vaguely explained and differs from panel to panel, which hampers the interpretation of the data. In addition, the effects are very slight and although can be described with p values the overall impact must be very small. There are many other major points that need to be addressed and this data is too preliminary for publication.

Major comments

Figure 1. A time course should be performed. Are the EEA1 and Rab11 levels still increased after 2h Starvation for instance? It is crucial to assess at which time point the levels of the 2 proteins start to decrease and whether this decrease is happening in a consistent way between Rab11 and EEA1 or if the kinetics/dynamics of their degradation during autophagy induction is different. Why are EEA1 and Rab11 levels increasing upon 1 hour starvation? This result is not discussed at all in the manuscript. How do the authors identify the tubulated endosomes from the vesicular portion? There is no mention of the methods they have used (not even in the Materials and Methods section). This must be added to determine how robust is their analysis. The changes in EEA1 morphology persist in short and long starvation (1/6h) suggesting that the differences in protein amount and endosome morphology may not be connected. How do the authors explain this difference? In Fig 1D. Image quantification. What are the parameters used for this analysis? What do the authors mean by non-vesicular EEA1? Based on the images all EEA1 analysed would correspond to membrane-bound EEA1. Can the authors show the analysis of circularity?

Figure 2. Is the EEA1 compartment tubulation that occurs during starvation increased even more upon SNX1 over-expression? Fig 2D/E/F. What is the mechanism proposed by the authors of tubulation at such early time points? The authors should discuss at some point how these changes might occur in such a short timepoints. Could this effect be mTORC1 dependent? Do endosomes tubulate upon Rapamycin treatment? Do these changes at 1 minute also correlate with an increased expression of the proteins involved (endogenous levels or even overexpressed SNX1)?

Figure 3. The authors performed LC3II analysis in cells depleted or not for SNX1. However, it is not clear whether in ctrl cells the difference between starvation and starvation plus bafilomycin is significant (as it should be in these conditions).

Figure 4. The authors show that among all the SNX1 positive tubular network, only few SNX1-positive tubules transiently interact with DFCP1-positive membrane structures that are enriched in PI3P. In Supp Fig 4, the authors show that the R186A-SNX1 PI3P binding mutant does not show the tubular network as SNX1 WT.

Thus, it is not clear why the interaction of SNX1 with PI3P is functional for the formation of its tubular network. Can the authors elaborate more on this point?

Was this mutant previously reported? The authors should state that. If the mutant is being reported here for the first time, the controls for PI3P binding should be included.

From the quantifications shown, it seems the mutant has a dominant negative effect on the endogenous SNX1 as it completely abolished LC3 puncta formation upon starvation. How the authors explain this effect? Has this effect previously reported for SNX1?

Figure 5. Panels 5C and D there is no mention about how the authors quantify the heterogeneity of SNX2 distribution on SNX1 tubules. The details of the analysis must be clarified.

The authors perform the analysis at very early time points, what happens at the time points where the rest of the effects are observed? Are these differences persistent? Add 1h and 6h to the analysis.

Panels 5F and G the criteria for the quantification are very unclear. The authors claim that for their quantification they have used the formula in fig. 5F, but the formula per se is lacking any details for the quantification analysis. The authors should elaborate much better how this quantification was performed?

The authors claim that "...in the timeframe of autophagosome biogenesis that we report to be associated with massive SNX1 endosomal tubulation (Figure 2), the interaction between endosomal SNX2 and VAPB is transiently stabilized". Considering that SNX1-SNX2 can heterodimerize, what happens to SNX2-VAPB interaction in SNX1 KD cells? Is this interaction reduced or abolished if SNX1 is functioning as platform for the transient assembly of SNX2-VAPB complex? Also is SNX1 KD affecting SNX2 endogenous levels (and the other way around)?

Figure 6. How is stable codistribution is defined? Which criteria were used for this analysis?

Figure 7. Using SNX2 KD cells, what happens in these conditions to the EEA1 positive compartment, is the starvation induced tubulation of this compartment blocked/impaired as in the case of SNX1?

What happens to the autophagic flux upon SNX2 siRNA?

General points

A missing requirement in the study is the analysis of early autophagic structures alongside with LC3 quantification. The study requires markers for all the analysed effects on autophagy to show defects in early autophagosome formation (WIPI2, DFCP, ULK1...) as well as the responsiveness of the signalling cascade when SNX1 is not there. Is mTORC1 equally inactivated and ULK1 operating as expected? In other words, where is the defect in autophagy starting? The study correlates SNX1 but the mechanisms is not clear.

The image quantification used in this manuscript is quite elaborate, but the criteria used to define the parameters as well as the justification of the quantification methods is very unclear, and it completely affects the interpretation of the results and is sometimes misleading. This must be carefully addressed throughout the manuscript to improve the quality.

In the discussion, the authors claim that SNX2 bridge between the endosomal compartment positive for SNX1 and the ER compartment positive for VAPB. Considering their data on SNX2-VAPB interaction as well as the fact that SNX1 and SNX2 can heterodimerize, could then they speculate the formation of a protein complex SNX1-SNX2-VAPB? Can they detect the interaction (undirect) between SNX1 and VAPB and is this interaction abolished upon SNX2 KD?

On the same line, in the discussion the authors point out the formation of this crosstalk between SNX2 and VAPB on SNX1 positive tubules. However, these tubulation events occurs artificially upon SNX1 overexpression. How do the authors justify such a model considering the endogenous SNX1?

In general terms, the discussion of this manuscript is somehow unfocused and does not put the results of the paper in context but rather presents a compendium of recent literature on autophagosome formation and it would be better to elaborate more the results presented in the manuscript throughout the discussion.

Reviewer #3 Review

Comments to the Authors (Required):

This manuscript investigates the involvement and contribution of early endosome-associated SNX1/SNX2 in autophagy. Authors show a critical involvement of SNX1 in starvation-induced autophagy by its loss or gain of function experiments. Mechanistically, SNX1-positive early endosomes become tubulated upon starvation and then merge with the DCFP1- positive ER domain where the autophagosome formation is supposed to take place, suggesting the formation of membrane contacts between SNX1-positive endosomes and the ER. SNX2, a binding partner of SNX1, is proposed to mediate the contact formation based on the data indicating an increase in its connection with VAPB morphologically (colocalization) as well as biochemically (coprecipitation) upon starvation. Knock down of SNX2 leads to a reduction of SNX1 association with the DCFP1-positive ER domain, further suggesting a role of SNX2 as the mediator of the ER-endosome contacts. Thus, authors propose that nutrient deprivation induces SNX1-mediated endosomal tubulation and subsequent contact formation with the ER via SNX2 and VAPB, thereby providing the platforms for autophagosome biogenesis.

This manuscript describes potentially interesting observations (i.e. morphological changes of early endosomes in a short starvation). However, some data are still preliminary, and their conclusions are not clearly supported by the data provided. Here are the comments/suggestions that would strengthen the current manuscript.

1. There is no direct evidence that membrane contacts between the SNX1/SNX2-endosomes and the ER are indeed the platforms for autophagosome formation. This is a major concern that should be clearly demonstrated in the current study to fully support the authors conclusion.
2. Another concern is that evidence for the membrane contacts between SNX1/SNX2 and VAPB (e.g. Fig.6) is rather weak. First of all, I would strongly encourage authors to provide solid evidence using other methods such as EM or CLEM. Second, authors should analyze the contacts located at more peripheral area that contains reticular ER tubules rather than perinuclear region shown in Fig 6C and 6D that contains highly crowded ER membranes.
3. To get more mechanistic insight into how SNX1 and SNX2 regulate endosomal mobilization and contact formation, a series of rescue experiments need to be done. For example, rescue by SNX2 wild-type and mutants (PX-mutants, BAR domain mutants) should be done in Fig. 7E. Authors also should provide rescue data for Fig.2.
4. SNX1 quickly responds to starvation, and this would be a quite interesting observation. Thus, live imaging of SNX1 as well as SNX2 during nutrient deprivation (before and after starvation) should be demonstrated to describe more precisely and understand deeply the dynamic behavior of SNX1 and SNX2 in response to starvation.
5. How about the levels of phosphoinositides such as PI3P and PI4P at the SNX1/SNX2-positive endosomes? Localization of PI3P or PI4P at those endosomes should be examined. Additionally, authors should test if the amount of PI3P or PI4P at the SNX1/SNX2-positive endosomes changes upon short starvation.

August 4, 2022

Re: Life Science Alliance manuscript #LSA-2022-01652-T

Dr. Etienne Morel
INEM, Paris
Cell Biology
14 rue Maria Helena Viera Da Silva
INSERM U1151-CNRS UMR 8253 Université Paris Descartes-Sorbonne Paris Cité
PARIS 75014
France

Dear Dr. Morel,

Thank you for submitting your manuscript entitled "A SNX1-SNX2-VAPB partnership regulates endosomal membrane rewiring in response to nutritional stress" to Life Science Alliance. We invite you to submit a revised manuscript addressing the following Reviewer comments:

- Address Reviewer 1's comments.
- Address Reviewer 2, with the comments about Figure 2 D/E/F, Figure 4 and Figure 5 addressed via Discussion. The questions about the formation of a protein complex SNX1-SNX2-VAPB should be addressed only if data is readily available.
- Address Reviewer 3's comments. Points #1, 2 and 4 can be addressed experimentally, or by toning down those conclusions.

Thank you for this interesting contribution to Life Science Alliance. We are looking forward to receiving your revised manuscript.

Sincerely,

B. MANUSCRIPT ORGANIZATION AND FORMATTING:

Life Science Alliance editor's comments:

- Address Reviewer 1's comments.
- Address Reviewer 2, with the comments about Figure 2 D/E/F, Figure 4 and Figure 5 addressed via Discussion. The questions about the formation of a protein complex SNX1-SNX2-VAPB should be addressed only if data is readily available.
- Address Reviewer 3's comments. Points #1, 2 and 4 can be addressed experimentally, or by toning down those conclusions.

Reviewer #1 (Comments to the Authors (Required)):

This study by De Graca et al. focuses on the possible role of SNX1 (and SNX2) during autophagy. This follows recent studies highlighting how interference with SNXs (e.g., SNX18; SNX4; SNX7) alters the starvation autophagy response, typically through ATG9A trafficking defects. Here the authors firstly address the effects of starvation on endosomal behavior, with tubulation of overexpressed SNX1-GFP observed, and apparent changes in EEA1 morphology. The study then goes on to address this with a series of mainly imaging-based experiments with some fairly limited autophagy assays that are restricted to LC3 puncta/lipidation analysis. Whilst there are some interesting initial findings regarding changes in endosomal behavior and direct or indirect effects on autophagy, the mechanistic side of the study is weak and relies heavily on quite subjective assessment of subtle changes in marker localization. The study is also severely weakened by the use of a single cell-line (HeLa) and single siRNAs with limited rescue experiments. In sum, the work is too preliminary and not convincing enough to be published in this journal.

We thank reviewer 1 for finding that some of the data in our initial manuscript were interesting. As described below, we now strengthened our conclusions by dedicated experiments that include, as suggested, testing our model in another cell line, rescue experiments and additional experiments addressing the mechanistic connecting endosomes and ER membranes in response to starvation. We notably re-enforced our data by adding more pre-autophagic markers readouts to several sets of analysis, as suggested by reviewer 1. Importantly, we now show that a rescue of SNX2 knockdown using a SNX2 mutant lacking VAPB interaction residue (F28A) is not able to drive the SNX1 endosomal tubule to DFCP1 puncta/site of pre-autophagosome assembly (see Figure 7E, 7F and 7G), strengthening our conclusions regarding the role of SNX1/SNX2/VAPB interplay in response to nutritional stress.

1. T-tests should not be used for n>2 comparisons.

R1.1. We thank this reviewer 1 for pointing out this inaccuracy. All legends (and/or main text) have been checked out and no t-test was used for n>2 comparisons in the revised version of the manuscript.

2. What is "endosomal membrane rewiring towards ER sub-domains"? Abstract

R1.2. We now reformulated the sentence in the revised abstract as follows: “we report that starvation mobilizes early endosome mobilization and induces tethering of endosomal tubules to ER sub-regions fostering phagophore assembly”.

3. Fig. 1A: transient increase in EEA1 and Rab11 needs further validation. Is this transcriptionally controlled? Does this involve decreased proteasomal (or even, although unlikely, autophagy)? Normalized against actin, could it not be possible that actin is transiently decreased? In other words, the choice of normalization could lead to anomalies here.

R1.3. We agree with reviewer 1 that these data might be misleading and removed them from the revised version of the manuscript.

4. EEA1 morphology experiment (Fig. 1C, D) and accompanying SNX1 analysis in Fig. S1 are weak. What is being analyzed in Fig. 1C? Is this measured by eye or has software been used for an objective readout? Why is the analysis in Fig. S1C, D extended out to 24h which is extremely late for starvation analysis?

R1.4. Reviewer 1 raises an important point and we have now explained the morphology quantification methods with clarifying details all through the revised version of the manuscript. Importantly, data shown in Figure 1 have been obtained with adapted parameters to tackle the alteration of endosomal morphology with better parameters, using image-analysis Software (ICY software using ROIs analyses). EEA1-endosome morphology has been quantified using two key factors of organelle morphological changes, namely the “roundness” (a normalized ratio between the radius of the minimum inscribed and largest circumscribed circle and expressed as a percentage), as shown in new Figure 1B, and the “elongation” (a scale factor corresponding to the ratio between the first and the second ellipse diameters where a non-elongated object has a minimal value of 1), as shown in new Figure 1C. As shown in Figure 1, starvation induces a decrease of EEA1-positive endosomes roundness and an increase of their elongation factor. The analysis in the supplementary figure 1C and 1D is ranging from 6h to 24h to show the evolution of SNX1-EEA1 codistribution phenotype in time.

5. Tubulation of SNX1-GFP tubules is interesting (Fig. 2), but could be forced by overexpression. In any case, this needs to be validated in a different cell-line(s). The accompanying data in Fig. S2 that attempts to demonstrate a role for SNX1 in early endosomal tubulation are poor - the accompanying images of EEA1 do not show any differences in tubular structures and thus appears to be highly subjective.

R1.5. Initial data concerning the modification of EEA1-endosomes morphology have been removed. We agree that SNX1-GFP transfection could lead to artificial tubulation of endosomes, although it remains so far the only available technique to visualize SNX1-positive organelle by live microscopy, since tubular structures are difficult to identify on fixed samples treated for immunohistochemistry. We now show that, as mentioned by reviewer 1, although SNX1-GFP transfection alters EEA1 endosome roundness, this

phenomenon is significantly amplified by starvation (new supplementary figure 2A and 2B), arguing for a *bona-fide* effect of nutrient deprivation on early endosome morphology.

Moreover, to strengthen our findings, we have analyzed live-imaging recordings during short time starvation or Rapamycin treatment (*i.e.* 2 and 5min) from a determined acquisition field. We show the tubulation of SNX1GFP structures during the time course of the experiment (new supplementary figure 2C and 2D (starvation) and 2G and 2H (Rapamycin)). Finally, and following reviewer 1 suggestions, we recapitulated our experiment in MEFs – mouse embryonic fibroblasts as shown in the new supplementary figure 2E and 2F, thus validating our data in a different cell line.

6. The autophagy response in siSNX1 cells does appear to be blunted. However, only one siRNA is used and there does not appear to be any attempt to rescue this effect. I would also prefer to see complete immunoblots in place of the montage shown in Fig. 3A.

R1.6. We apologize for the insufficient information regarding the systematic double siRNA use in the previous version of our manuscript. We indeed systematically used 2 validated siRNA oligos for each experiments requiring transfection-mediated gene downregulation. We also performed, as requested by reviewer, a rescue experiment using SNX1-GFP overexpression in cells knockdown for endogenous SNX1. Our results show that exogenous SNX1 boosts basal autophagy (as reported in supplementary Figure 4), and rescues the absence of endogenous SNX1 during autophagy (see figure R1.6 below).

[Figure has been removed by LSA Editorial Staff per authors' request.]

7. The overexpression effect looks convincing (Fig. S3), but its biological significance will always be questionable. The attempt to confirm the relevance of this through use of the R186A mutant (Fig. S4) throws up more questions than it answers because of unusual effects on total LC3 levels. What do the authors suggest is happening here? Is there a dominant negative effect, and if so what is the mechanism?

R1.7. Data initially presented in Fig S4 have been removed from this updated version of the manuscript to focus the message of the paper.

8. I was not convinced by the co-localization analysis between SNX1 and DFCP1 (Fig. 4) - since the distribution of SNX1 (punctate to tubular) is a feature of the starvation response, chance colocalization is likely to increase. A further control is needed. For the SNX1-LC3 colocalization, this appears more likely, but it should be noted that RFP-LC3 will still be fluorescent in the lysosome so we may be looking at late autophagosome/lysosomal colocalization here, which is not consistent with a role for SNX1 during early autophagosome biogenesis.

R1.8. Following the suggestion of reviewer 1 we performed similar experiments (and analysis/workflow/quantifications) using other markers of the autophagy pathway, not related to autophagosome biogenesis. Data presented in new supplementary Figure 6E, 6F and 6G show that codistribution of SNX1 with LAMP2A , a marker of late endosome/lysosome, was not increased upon starvation, strengthening our data concerning the detection of SNX1 positive membranes at the vicinity of ER-based autophagosome biogenesis platform in response to nutritional stress.

We now describe better the image analysis processes used in Figure 4 (B, C and D, for DFCP1) and supplementary Figure 6 (B, C and D, for LC3). As detailed in the supplementary Figure 5, the 3D surface-rendering workflow we set up to analyze the vicinity of SNX1 (using antibody) and DFCP1/Sec61b (transfected) was based on fixed samples and thus the SNX1 tubules were undetectable. Sustained by computer and machine learning analyses, creation of binary masks and surface-rendering allowed us to quantify events of SNX1 codistribution with DFCP1 (a marker of pre-autophagic structures) at ER subdomains and to compare them in control and short-time starvation points.

9. The data in Fig. 5 are interpreted as meaning that SNX1 establishes tubules during starvation along which SNX2 foci can be found, and that these have a tendency to co-distribute with LC3. This is potentially interesting but needs more work to validate e.g., other autophagy markers for early stages (e.g., WIPI2; ATG16L1), higher resolution microscopy including correlative light-electron microscopy and immunoEM.

R1.9. Following reviewer 1 suggestions, we performed similar experiments using ATG16L1, a marker of early stages of autophagy. As displayed in the new supplementary figure 7, similarly to what we reported in Figure 5E, F and G, SNX2-associated fluorescence along SNX1 tubule was systematically increased when ATG16L1 was detected at the vicinity of SNX2.

10. Fig. 6, the VAPB colocalization is unconvincing and the quantitation shown in Fig. 6E is poorly defined and makes no sense to me. The IP experiment in Fig. 6F, G is potentially interesting but needs to be shown using native proteins.

R1.10. We removed the data (initially presented in Fig 6E) related to 3D live imaging in the revised version of the manuscript. However, to re-enforce the exploration of SNX1 tubules/VAPB interplay, we analyzed the codistribution of SNX1 and VAPB with SNX2, ATG16L1 and WIPI2 (marker of early autophagy stages). Data displayed in Figure 6B to G

show that SNX2 presence on SNX1 tubules correlates with the presence of VAPB (Figure 6B and 6C) and that VAPB codistribution with SNX1 is associated with the presence of early autophagic markers such as ATG16L1 and WIPI2 (Figure 6D to 6G).

Concerning the co-IP experiments we wish to precise that the initial discovery and analysis of VAPB-SNX2 interaction was made using overexpression of both partners (Dong et al. 2016, Cell). Here (Figure 6H and 6I) we used SNX2-YFP as a bait (and SNX3-GFP as a negative control) but we detected endogenous VAPB from the co-immunoprecipitate.

11. In Fig. 7 a single siRNA is used to deplete SNX2 to assess the behavior of overexpressed SNX1-GFP. It looks as though SNX1-GFP is already more tubulated before starvation (Fig. 7C), meaning that the analysis as presented in Fig. 7D misses that effect. Fig. 7E, F appears to have been analyzed by eye-balling apparent co-distribution. If so, this lacks objectivity.

R1.11. Again, and as mentioned in point R1.6, we do apologize for the lack of details concerning siRNA's use in the previous version of our manuscript. 2 different siRNAs were systematically used in experiments requiring gene expression downregulation, including in Figure 7 (as in new supplementary figure 9).

Concerning the quantification in Figure 7D, the variation data show that, although – and as mentioned by this reviewer – SNX1 positive endosome tend to tubulate more in the absence of SNX2, this tubulation is not increased in starvation situation, as shown in siCTRL cells (see Figure for reviewer#2 R2.12C and D).

Moreover, we would like to point out that the new experiments presented in Figure 7F and 7G show that SNX1 codistribution with DFCP1 is altered in the absence of SNX2 (siSNX2) and that a VAPB-binding defective mutant of SNX2 (F28A SNX2) is not able to rescue this process. All the information concerning image acquisition and quantification are now detailed in the revised version of the manuscript.

Reviewer #2 (Comments to the Authors (Required)):

This paper describes a role for SNX1 in the tubulation of early endosomal compartment upon nutrient starvation. The authors propose that the SNX1-positive tubulated endosomes can interact through SNX2 with ER subdomains positive for VAPB associated with phagophore biogenesis. The identification of the endosomal compartment as a membrane reservoir for early autophagy is not novel as membrane contributions have been shown to occur from recycling endosomes via Snx18 (Knævelsrud et al. JCB 2013). The authors use a combination of microscopy with image quantification and analysis to support their model and use biochemistry to assess autophagic flux and protein interactions. However, despite the identification of the role of SNX1-SNX2-VAPB as main actors the molecular details of the interaction of the three proteins is not clear. Nor is sufficient detail provided on how each component influences the others in the assembly of this protein/membrane complex upon starvation. Even if the imaging in this paper is of high quality, the methodology used for the analysis of the data is many times vaguely explained and differs from panel to panel, which hampers the interpretation of the data. In addition, the effects are very slight and although can be described with p values the overall impact must be very small. There are many other major points that need to be addressed and this data is too preliminary for publication.

Major comments

Figure 1. A time course should be performed. Are the EEA1 and Rab11 levels still increased after 2h Starvation for instance? It is crucial to assess at which time point the levels of the 2 proteins start to decrease and whether this decrease is happening in a consistent way between Rab11 and EEA1 or if the kinetics/dynamics of their degradation during autophagy induction is different.

Why are EEA1 and Rab11 levels increasing upon 1 hour starvation? This result is not discussed at all in the manuscript.

R2.1. As mentioned by reviewer 2 (see below), the differences in protein amount and endosome morphology may not be directly connected. Initial data concerning variation of EEA1-endosomes morphology have thus been removed from the present version of the manuscript.

How do the authors identify the tubulated endosomes from the vesicular portion? There is no mention of the methods they have used (not even in the Materials and Methods section). This must be added to determine how robust is their analysis.

The changes in EEA1 morphology persist in short and long starvation (1/6h) suggesting that the differences in protein amount and endosome morphology may not be connected. How do the authors explain this difference?

In Fig 1D. Image quantification. What are the parameters used for this analysis? What do the authors mean by non-vesicular EEA1? Based on the images all EEA1 analysed would correspond to membrane-bound EEA1. Can the authors show the analysis of circularity?

R2.2. We thank reviewer 2 for his comment which led us to better describe the morphology quantification methods all through the revised version of the manuscript.

Moreover, as suggested by this reviewer, we now robustly analyzed the alteration of endosomal morphology with new parameters, using an image-analysis Software (Icy using ROIs analyses). EEA1-endosome morphology was quantified using two key factors of organelle morphological changes, namely the “roundness” (a normalized ratio between the radius of the minimum inscribed and largest circumscribed circle, expressed as a percentage), as shown in new Figure 1B, and the “elongation” (a scale factor given by the ratio between the first and the second ellipse diameters where a non-elongated object has a minimal value of 1), as shown in new Figure 1C. As shown in Figure 1, starvation induced a decrease of EEA1-positive endosomes roundness and an increase of their elongation factor.

Figure 2. Is the EEA1 compartment tubulation that occurs during starvation increased even more upon SNX1 over-expression?

R2.3. Initial data concerning variation of EEA1-endosomes morphology have been removed and we agree that SNX1-GFP transfection could boost artificially tubulation of endosomes, although it remains so far the only available technique to visualize SNX1-positive organelle by live microscopy. We now show that, as suggested by reviewer 2, although SNX1-GFP transfection indeed alters EEA1 endosome roundness (and thus tubulation), this phenomenon is significantly amplified by starvation (new supplementary figure 2A and 2B), arguing for a *bona-fide* effect of nutrient deprivation on early endosome morphology.

Fig 2D/E/F. What is the mechanism proposed by the authors of tubulation at such early time points? The authors should discuss at some point how these changes might occur in such a short timepoints. Could this effect be mTORC1 dependent? Do endosomes tubulate upon Rapamycin treatment? Do this changes at 1 minute also correlate with an increased expression of the proteins involved (endogenous levels or even overexpressed SNX1)?

R2.4. Following reviewer 2 recommendations, we now discuss more extensively the tubulation of endosomal membranes upon starvation. To further re-enforce these data, and in accordance with the reviewer’s suggestion, we have analyzed live-imaging data performed during short time starvation or Rapamycin (*i.e.* 2 and 5min) from a determined acquisition field and we show tubulation of SNX1GFP structures during the time course of the experiment (new supplementary figure 2C and 2D (starvation) and 2G and 2H (Rapamycin)). Finally, and following reviewer #1 suggestions, we performed similar experiments in a different cell line (MEFs – mouse embryonic fibroblasts) as shown in the new supplementary figure 2E and 2F.

Figure 3. The authors performed LC3II analysis in cells depleted or not for SNX1. However, it is not clear whether in ctrl cells the difference between starvation and starvation plus bafilomycin is significant (as it should be in these conditions).

R2.5. We understand the point of view of the reviewer. In the initial version of the manuscript, the goal of the experiment was not to assess the autophagic flux *per se*, but to compare the autophagic response between the siCTRL and the siSNX1 cells. We have now modified the organization of the data and new Figure 3A, 3B, and supplementary figure 3, show experiment using BafA1 to block autophagic flux. As described in Figure 3A and 3B, the number of LC3 puncta is reduced both in presence of absence of BafA1 and, as seen in

supplementary figure 3C and 3D, the lipidation of LC3 is reduced with and without BafA1, indicating that the autophagic flux is altered in absence of SNX1.

Importantly, we also added new autophagic readouts to analyze the effects of SNX1 depletion on the early stages of autophagy. In new Figure 3C, 3D and 3E we now show that the total number of ATG16L1 structures as well as its colocalization with WIPI2 (considered as *bona-fide* readout for autophagosome biogenesis) is reduced in siSNX1 transfected cells, supporting the hypothesis that the early steps of the autophagosome assembly sequence are indeed compromised in this situation.

Figure 4. the authors show that among all the SNX1 positive tubular network, only few SNX1-positive tubules transiently interact with DFCP1-positive membrane structures that are enriched in PI3P. In Supp Fig 4, the authors show that the R186A-SNX1 PI3P binding mutant does not show the tubular network as SNX1 WT.

Thus, it is not clear why the interaction of SNX1 with PI3P is functional for the formation of its tubular network. Can the authors elaborate more on this point? Was this mutant previously reported? The authors should state that. If the mutant is being reported here for the first time, the controls for PI3P binding should be included.

From the quantifications shown, it seems the mutant has a dominant negative effect on the endogenous SNX1 as it completely abolished LC3 puncta formation upon starvation. How the authors explain this effect? Has this effect previously reported for SNX1?

R2.6. To maintain the focus on the manuscript core messages, data about PI3P and R186A-SNX1 mutant were removed from this updated version. Reviewer 2 pinpoints an important question since both SNX1 and DFCP1 are indeed PI3P-binding proteins and PI3P can be found at endosomal and pre-autophagosomal/(ER-associated) omegasome membranes (when autophagy is induced). From what we know, SNX1 only associates with endosomal membrane (via its PX domain and endosomal PI3P platforms (PMID: 11997453, 12198132 and 15673616) and DFCP1 associates with ER-omegasome. This raises up the question of putative PI3P transfer/exchange from one compartment to another during autophagosome biogenesis, which we mention in the discussion section of the paper.

Figure 5. Panels 5C and D: there is no mention about how the authors quantify the heterogeneity of SNX2 distribution on SNX1 tubules. The details of the analysis must be clarified.

R2.7. We thank reviewer 2 for pointing out this omission. We modified the methods section accordingly. In brief, using live imaging acquisitions, we quantified the number of SNX1-RFP tubules which did not display a complete colocalization (along the tubule) with SNX2-YFP, therefore terming them “heterogeneous for SNX2”.

The authors perform the analysis at very early time points, what happens at the time points where the rest of the effects are observed? Are these differences persistent? Add 1h and 6h to the analysis.

R2.8. Data shown in Figure 5A and 5B illustrate an alteration in SNX1/SNX2 codistribution in response to 1h of starvation. However, as mentioned in the manuscript (see Figure 2), the use of antibodies in fixed cells limits the detection of fragile and long endosomal tubules. Experiments shown in Figure 5C and 5D derive from live-cell imaging using SNX1RFP and SNX2YFP, and were analyzed to decipher the dynamics of SNX1/SNX2 codistribution upon very early time points after starvation (i.e. 2min and 10min) because combining GFP tagged expression and live imaging favors detection of endosomal tubules. Such experiment is particularly difficult to achieve with 1h and 6h of starvation, because of the limitations of live imaging. However, and to precisely address this point, we would like to highlight that similar experiment were made after 1h of starvation - in fixed cells expressing SNX1GFP – and are shown in Figure 5E (SNX1GFP, SNX2 and LC3), in new Figure 6B (SNX1GFP, SNX2, VAPB) and in new supplementary Figure 7 (SNX1GFP, SNX2 and ATG16L1). In these settings, we clearly see that SNX2 is no longer present all along the SNX1 tubule (as observed in Figure 5C (panel ctrl), or in Figure 5A (panel ctrl)) but segregates at specific subdomains of the SNX1 endosomal structures that are enriched with LC3 (Figure 5E, 5F and 5G), ATG16L1 (supplementary Figure 7) and VAPB (Figure 6B and 6C).

Panels 5F and G the criteria for the quantification are very unclear. The authors claim that for their quantification they have used the formula in fig. 5F, but the formula per se is lacking any details for the quantification analysis. The authors should elaborate much better how this quantification was performed?

R2.9. Again, we apologize for every omission concerning the description of the quantification methods. We modified the methods section accordingly. To address the correlation between the presence of SNX2 along the SNX1 tubule and its codistribution with LC3 signal we analyzed the RGB plot of all 3 channels along the SNX1-GFP tubule (using Fiji software and the «Color Profiler.jar» plug-in made by Dimiter Prodanov, see <https://imagej.nih.gov/ij/plugins/color-profiler.html>). On the RGB plot graph we measured the spot of maximal and minimal intensity for LC3 (Y axes). Keeping the defined X coordinates of these 2 points we measured the fluorescence intensity of the SNX2 signals. Both measurements were subtracted from the minimal SNX2 intensity within the same RGB plot to normalize all quantifications.

The authors claim that "...in the timeframe of autophagosome biogenesis that we report to be associated with massive SNX1 endosomal tubulation (Figure 2), the interaction between endosomal SNX2 and VAPB is transiently stabilized". Considering that SNX1-SNX2 can heterodimerize, what happen to SNX2-VABP interaction in SNX1 KD cells? Is this interaction reduced or abolished if SNX1 is functioning as platform for the transient assembly of SNX2-VAPB complex? Also is SNX1 KD affecting SNX2 endogenous levels (and the other way around)?

R2.10. Reviewer 2 raised an important point. We addressed his question by performing co-immunoprecipitation experiments in cells expressing SNX2YFP and knocked down for SNX1. As shown below (see R2.10 figure) the amount of SNX2YFP-immunoprecipitated VAPB is lower in siSNX1 cells, notably in starvation conditions. This suggests, as pointed out by this reviewer, that SNX1 participates, at least partially in the transient assembly of SNX2 and VAPB.

[Figure has been removed by LSA Editorial Staff per authors' request.]

Moreover, following the suggestion of reviewer 2, we tested the effects of SNX1 knockdown (KD) on SNX2 stability (and the other way around). As shown in the new supplementary Figure 9, we now report that neither SNX1 nor SNX2 KD have incidence on SNX2 and SNX1 stability, respectively. This suggests that despite their ability to heterodimerize, SNX1 and SNX2 might endorse specific biological functions in specific biological situations, such as upon response to starvation, as described in the present manuscript.

Figure 6. How is stable codistribution is defined? Which criteria were used for this analysis?

R2.11. We apologize for the lack of clarity in the technical description of this experiment. Because we were using 3D live imaging, we were able to track endosomal SNX1 positive codistribution with VAPB domains in time. However, to gain in clarity and focus on the manuscript core messages, these have been removed from the present version of the paper.

Figure 7. Using SNX2 KD cells, what happen in these conditions to the EEA1 positive compartment, is the starvation induced tubulation of this compartment blocked/impaired as in the case of SNX1? What happens to the autophagic flux upon SNX2 siRNA?

R2.12. Following the question of reviewer 2, we analyzed the EEA1 endosomal compartment dynamics and the autophagic flux in response to starvation, in cells depleted for SNX2. We show that while the number of LC3 positive dots is unchanged in basal conditions, the number of LC3 structures is significantly reduced (in presence and in absence of BafA1) in siSNX2 cells (Figure R2.12 A and B, see below). Moreover, using the “roundness” parameter (to analyze endosome morphology, see Figure 1), we show that EEA1 endosomes were less circular in cells lacking SNX2, but that these endosomes were not further modified (as observed in control cells, see Figure 1) upon starvation (Figure R2.12. C and D, see below) indicating that SNX2 is required for endosomal morphodynamics modification in response to starvation.

[Figure has been removed by LSA Editorial Staff per authors' request.]

General points

A missing requirement in the study is the analysis of early autophagic structures alongside with LC3 quantification. The study requires markers for all the analysed effects on autophagy to show defects in early autophagosome formation (WIPI2, DFCP, ULK1...) as well as the responsiveness of the signalling cascade when SNX1 is not there.

R2.13. as suggested by this reviewer and by reviewer 1, we now performed a new set of data using pre-autophagic markers such as WIPI2 and ATG16L1 (Figure 3, Figure 6 and supplementary Figure 7) that re-enforce data obtained with LC3 (Figure 3, Figure 5, supplementary Figure 4 and supplementary Figure 6) and DFCP1 (Figure 4 and Figure 7).

Is mTORC1 equally inactivated and ULK1 operating as expected? In other words, where is the defect in autophagy starting? The study correlates SNX1 but the mechanisms is not clear.

R2.14. Data shown in supplementary figure 2G and 2H show that SNX1 endosomal tubulation also occur in response to Rapamycin treatment, indicating that this process is associated with mTOR inhibition. Whether the mTOR/ULK1 complex is affected or not in cells KD for SNX1 is an important point, but the main focus of the present manuscript being about endosomal morphodynamics in response to starvation, we do believe that this point is out of the scope of the paper.

The image quantification used in this manuscript is quite elaborate, but the criteria used to define the parameters as well as the justification of the quantification methods is very unclear, and it completely affects the interpretation of the results and is sometimes misleading. This must be carefully addressed throughout the manuscript to improve the quality.

R2.15. As mentioned earlier, we fully agreed with this reviewer's remark: the previous version of our manuscript lacked the necessary precision for imaging quantifications. We corrected the present version accordingly.

In the discussion, the authors claim that SNX2 bridge between the endosomal compartment positive for SNX1 and the ER compartment positive for VAPB. Considering their data on SNX2-VAPB interaction as well as the fact that SNX1 and SNX2 can heterodimerize, could then they speculate the formation of a protein complex SNX1-SNX2-VAPB? Can they detect the interaction (undirect) between SNX1 and VAPB and is this interaction abolished upon SNX2 KD?

R2.16. This is indeed an interesting point. However, from our own experiments (not shown) strong co-immunoprecipitation were found for SNX1 with SNX2-YFP as bait, but this interaction did not change upon starvation treatment. Unfortunately, we were not able to find VAPB in co-immunoprecipitates when using SNX1-GFP as bait.

On the same line, in the discussion the authors point out the formation of this crosstalk between SNX2 and VAPB on SNX1 positive tubules. However, these tubulation events occurs artificially upon SNX1 overexpression. How do the authors justify such a model considering the endogenous SNX1?

R2.17. as mentioned in the R2.3 response to this reviewer we agree that SNX1-GFP transfection could lead to artificial tubulation of endosomes, although it remains so far the only available technique to visualize SNX1-positive organelle by live microscopy. We now show that EEA1 endosome roundness is decreased by starvation in SNX1 transfected cells as well (new supplementary figure 2A and 2B), arguing for a *bona-fide* effect of nutrient deprivation on early endosome morphology.

In general terms, the discussion of this manuscript is somehow unfocused and does not put the results of the paper in context but rather presents a compendium of recent literature on autophagosome formation and it would be better to elaborate more the results presented in the manuscript throughout the discussion.

R2.18. As suggested by this reviewer, we now focused more on the results in the revised discussion section of the manuscript.

Reviewer #3 (Comments to the Authors (Required)):

This manuscript investigates the involvement and contribution of early endosome-associated SNX1/SNX2 in autophagy. Authors show a critical involvement of SNX1 in starvation-induced autophagy by its loss or gain of function experiments. Mechanistically, SNX1-positive early endosomes become tubulated upon starvation and then merge with the DFCP1- positive ER domain where the autophagosome formation is supposed to take place, suggesting the formation of membrane contacts between SNX1-positive endosomes and the ER. SNX2, a binding partner of SNX1, is proposed to mediate the contact formation based on the data indicating an increase in its connection with VAPB morphologically (colocalization) as well as biochemically (coprecipitation) upon starvation. Knock down of SNX2 leads to a reduction of SNX1 association with the DFCP1-positive ER domain, further suggesting a role of SNX2 as the mediator of the ER-endosome contacts. Thus, authors propose that nutrient deprivation induces SNX1-mediated endosomal tubulation and subsequent contact formation with the ER via SNX2 and VAPB, thereby providing the platforms for autophagosome biogenesis.

This manuscript describes potentially interesting observations (i.e. morphological changes of early endosomes in a short starvation). However, some data are still preliminary, and their conclusions are not clearly supported by the data provided. Here are the comments/suggestions that would strengthen the current manuscript.

We thank this reviewer for finding that our work on morphological changes of endosomes during starvation describes potentially interesting observations and we thank her/him for providing helpful suggestions to strengthen the message of the manuscript.

1. There is no direct evidence that membrane contacts between the SNX1/SNX2-endosomes and the ER are indeed the platforms for autophagosome formation. This is a major concern that should be clearly demonstrated in the current study to fully support the authors conclusion.

R3.1. We apologize for the misunderstanding/overstatement. We were not expecting to say that membrane connections established through SNX1/SNX2 with ER-VAPB are only autophagosome formation spots. Indeed, our data do not show a direct membrane platform for autophagosome formation per se but rather the recruitment of the early autophagic machinery at the specific area of tethering. The putative direct contribution – and associated mechanism - of these membrane contacts, which we considered beyond the scope for this manuscript, is indeed very interesting and will be addressed in future studies.

VAPB has been itself described at being part of the ULK1/FIP200 complex during omegasome subdomain formation (PMID: 29628370) and our data (including new data) show that SNX1 tubule associated SNX2 puncta are not only enriched at VAPB subdomains, but also at LC3, ATG16L1 and WIPI2, which are also markers of early stages of autophagosome assembly sequence. These data are shown in Figures 3, 6 and supplementary 7 (for WIPI2 and ATG16L1) and supplementary Figure 4, Figure 4 and Figure 7 (for LC3 and DFCP1).

2. Another concern is that evidence for the membrane contacts between SNX1/SNX2 and VAPB (e.g. Fig.6) is rather weak. First of all, I would strongly encourage authors to provide

solid evidence using other methods such as EM or CLEM. Second, authors should analyze the contacts located at more peripheral area that contains reticular ER tubules rather than perinuclear region shown in Fig 6C and 6D that contains highly crowded ER membranes.

R3.2. We apologize for the lack of quality of this experiment, which addressed the analysis of immediate vicinity of SNX1 endosomes with VAPB ER subdomains. Because we were using 3D live imaging, we were able to track the SNX1 positive endosome stable connection to VAPB domains in time. However, to gain clarity and focus in the manuscript core messages, these have been removed from the present version of the paper. The data arguing for putative contact between SNX1 structures and ER subdomains (as shown in Figure 4 for DFCP1 and in supplementary Figure 6 for LC3, in live and 3D microscopy) have been re-enforced by new set of experiments and quantifications showing a specific localization of SNX2 on SNX1 tubules facing VAPB, WIPI2 and ATG16L1 (Figure 6B to 6G).

3. To get more mechanistic insight into how SNX1 and SNX2 regulate endosomal mobilization and contact formation, a series of rescue experiments need to be done. For example, rescue by SNX2 wild-type and mutants (PX-mutants, BAR domain mutants) should be done in Fig. 7E.

R3.3. We thank this reviewer for the helpful suggestion. Accordingly, we performed experiments using a mutant version of SNX2 unable to bind to VAPB (F28A, (PMID 27419871), as shown in new Figure 7E). Interestingly, we report that while endogenous SNX2 rescues the SNX1-DFCP1 codistribution phenotype that we show to occur in response to starvation (see Figure 4 and Figure 7F and 7G (siCTRL and siSNX2 + wtSNX2)), the F28A mutant failed to do so (Figure 7F and 7G (siSNX2 + F28ASNX2)). These data strengthen our conclusion about the role of SNX2 in driving the SNX1 endosomal structures to VAPB ER subdomains of the ER that support omegasome formation.

Authors also should provide rescue data for Fig.2.

R3.4. no siRNA/knockdown has been used in the Figure 2. However, we wish to highlight that such experiment (concerning SNX1 KD as shown in Figure 3 and supplementary Figure 3) has been done as suggested by reviewer #1 (see R1.6 Figure).

4. SNX1 quickly responds to starvation, and this would be a quite interesting observation. Thus, live imaging of SNX1 as well as SNX2 during nutrient deprivation (before and after starvation) should be demonstrated to describe more precisely and understand deeply the dynamic behavior of SNX1 and SNX2 in response to starvation.

R3.5. We thank this reviewer for this recommendation. We performed the suggested experiment in HeLa (and mouse embryonic fibroblasts) cells upon 2 and 5min of starvation and Rapamycin (new supplementary Figure 2C to H). These results describe evolution of given SNX1-positive endosomal structures (same cell, same acquisition fields) in response to starvation. Moreover, evolution of SNX2 vs SNX1 patterning in response to starvation is described in Figure 5.

5. How about the levels of phosphoinositides such as PI3P and PI4P at the SNX1/SNX2-positive endosomes? Localization of PI3P or PI4P at those endosomes should be examined. Additionally, authors should test if the amount of PI3P or PI4P at the SNX1/SNX2-positive endosomes changes upon short starvation.

R3.6. Although this is obviously a very interesting topic we do believe that the question of PI3P and PI4P turnover at surface of SNX1 endosomes is beyond the scope for the present paper. However, this aspect of endosomal membrane regulation is discussed in the revised version of the manuscript.

December 12, 2022

RE: Life Science Alliance Manuscript #LSA-2022-01652-TR

Dr. ETIENNE MOREL
Université Paris Cité
Cell Biology
Institut Necker Enfants Malades - INSERM U1151
156 Rue de Vaugirard
PARIS 75015
France

Dear Dr. Morel,

Thank you for submitting your revised manuscript entitled "a SNX1-SNX2-VAPB partnership regulates endosomal membrane rewiring in response to nutritional stress". We would be happy to publish your paper in Life Science Alliance pending final revisions necessary to meet our formatting guidelines.

- please use the [10 author names, et al.] format in your references (i.e. limit the author names to the first 10)
- please add the legend for your video to the main manuscript text

A. FINAL FILES:

B. MANUSCRIPT ORGANIZATION AND FORMATTING:

Sincerely,

Reviewer #1 (Comments to the Authors (Required)):

This is a revised version of the manuscript originally submitted to another journal. I have no further comments on the current manuscript.

Reviewer #2 (Comments to the Authors (Required)):

The authors have nicely addressed all my concerns in a substantially revised manuscript which is now suitable for publication.

December 19, 2022

RE: Life Science Alliance Manuscript #LSA-2022-01652-TRR

Dr. ETIENNE MOREL
Université Paris Cité
Cell Biology
Institut Necker Enfants Malades - INSERM U1151
156 Rue de Vaugirard
PARIS 75015
France

Dear Dr. Morel,

Thank you for submitting your Research Article entitled "a SNX1-SNX2-VAPB partnership regulates endosomal membrane rewiring in response to nutritional stress". It is a pleasure to let you know that your manuscript is now accepted for publication in Life Science Alliance. Congratulations on this interesting work.

DISTRIBUTION OF MATERIALS:

Again, congratulations on a very nice paper. I hope you found the review process to be constructive and are pleased with how the manuscript was handled editorially. We look forward to future exciting submissions from your lab.

Sincerely,
